# Structural basis for the recognition and degradation of host TRIM proteins by *Salmonella* effector SopA

Evgenij Fiskin[1,*], Sagar Bhogaraju[1,2,*], Lina Herhaus[1], Sissy Kalayil[1,2], Marcel Hahn[1] & Ivan Dikic[1,2,3]

The hallmark of *Salmonella* Typhimurium infection is an acute intestinal inflammatory response, which is mediated through the action of secreted bacterial effector proteins. The pro-inflammatory *Salmonella* effector SopA is a HECT-like E3 ligase, which was previously proposed to activate host RING ligases TRIM56 and TRIM65. Here we elucidate an inhibitory mechanism of TRIM56 and TRIM65 targeting by SopA. We present the crystal structure of SopA in complex with the RING domain of human TRIM56, revealing the atomic details of their interaction and the basis for SopA selectivity towards TRIM56 and TRIM65. Structure-guided biochemical analysis shows that SopA inhibits TRIM56 E3 ligase activity by occluding the E2-interacting surface of TRIM56. We further demonstrate that SopA ubiquitinates TRIM56 and TRIM65, resulting in their proteasomal degradation during infection. Our results provide the basis for how a bacterial HECT ligase blocks host RING ligases and exemplifies the multivalent power of bacterial effectors during infection.

[1] Institute of Biochemistry II, Goethe University School of Medicine, Theodor-Stern-Kai 7, 60590 Frankfurt am Main, Germany. [2] Buchmann Institute for Molecular Life Sciences, Goethe University, Max-von-Laue-Strasse 15, 60438 Frankfurt am Main, Germany. [3] Department of Immunology and Medical Genetics, School of Medicine, University of Split, Soltanska 2, 21000 Split, Croatia. * These authors contributed equally to this work. Correspondence and requests for materials should be addressed to I.D. (email: Ivan.Dikic@biochem2.de).

Salmonella enterica serovar Typhimurium is a Gram-negative pathogenic bacterium, which represents a major cause of food- and water-borne disease. Non-typhoidal Salmonella strains, including S. Typhimurium, cause severe gastroenteritis in immunocompetent individuals, whereas systemic infection can arise in immunosuppressed hosts. S. Typhimurium invasion and the concomitant induction of intestinal inflammation are initiated by secreted bacterial effector proteins, which are translocated into host cells via a channel-forming multi-protein complex known as the type-3 secretion system (T3SS)[1].

Bacterial infection is sensed by host pattern recognition receptor-mediated detection of pathogen-associated molecular patterns, such as lipopolysaccharide or bacteria-derived nucleic acids, and induces a pro-inflammatory state to combat infection[2]. The propagation of this innate immune response requires posttranslational modification of assembled receptor signalling complexes with ubiquitin (Ub)[3], which in eukaryotes is catalysed by three distinct classes of E3 Ub ligases known as homologous to E6–AP carboxy terminus (HECT), really interesting new gene (RING) and RING-between-RING.

To counteract Ub-dependent induction of host inflammatory signalling and microbicidal programmes, a wide range of bacteria have acquired strategies to subvert the host Ub proteasome system. Despite lacking the canonical Ub proteasome system, prokaryotic pathogens encode various families of virulence promoting E3 ligases. After their T3SS-mediated translocation, these ligase effectors hijack the host ubiquitination machinery and use their intrinsic catalytic activity to modify specific cellular targets[4–7]. The Salmonella T3SS effector protein SopA is a HECT-like E3 ligase that promotes Salmonella infection-induced inflammation[8–11]. Lacking noticeable primary sequence homology to eukaryotic HECT enzymes, the crystal structure of SopA showed the characteristic bi-lobal architecture of HECT ligases[9,12,13]. SopA-catalysed ubiquitination proceeds via a thioester-linked SopA∼Ub intermediate and requires an active site cysteine in its C terminus[8]. Subsequently solved structures of SopA in complex with E2 Ub-conjugating enzymes revealed an E2-binding site located within the N-lobe of SopA and further indicate a high structural flexibility of its C-lobe[14]. Structural characterization of SopA also uncovered the presence of an amino-terminal β-helix domain, whose function so far remains unknown[9].

Recent work implicates SopA in the regulation of two tripartite-motif containing (TRIM) E3 ligases TRIM56 and TRIM65 (ref. 15). TRIM proteins constitute a large family of ~70 RING-type E3 ligases, which plays a pivotal role in the host innate immune response against various pathogens[16,17]. TRIM56 and TRIM65 in particular have been demonstrated to stimulate type I interferon expression in conjunction with nucleic acid-sensing receptors such as STING, RIG-I and MDA5 (refs 15,18,19). TRIMs are characterized by the presence of three common structural features, consisting of an N-terminal catalytic RING domain, one or two B-boxes and a coiled-coil[20]. Multiple studies demonstrated that the coiled-coil of various TRIMs mediates anti-parallel dimer formation[21–24]. Furthermore, catalytic activity of TRIM proteins is enhanced by coiled-coil-dependent dimerization and, more recently, was shown to require RING domain dimerization for the proper activation of the E2∼Ub conjugate[25,26].

Here, using quantitative proteomics, structural and biochemical analysis we elucidate how Salmonella HECT-like ligase SopA specifically targets and inhibits human TRIM56 and TRIM65. We present the structure of the SopA–TRIM56 RING complex providing the molecular basis for SopA target specificity. Analysis of this complex further reveals that SopA occludes the E2 binding site of TRIM56 RING and inhibits TRIM ligase activity. Finally, we demonstrate that SopA directly mediates ubiquitination and proteasomal degradation of TRIM56 and TRIM65.

## Results

### Identification of TRIM56 and TRIM65 as SopA interactors.
Aiming to understand the molecular basis of SopA function during Salmonella pathogenesis, we set out to identify SopA host interactions using affinity purification coupled mass spectrometry (MS). To this end, we generated a stable HeLa Flp-In T-REx cell line inducibly expressing GFP-SopA upon the addition of doxycycline (Supplementary Fig. 1a). These cells were differentially labelled using stable isotope labelling by amino acids in cell culture (SILAC) and then either left untreated or treated with doxycycline to induce GFP-SopA expression. Next, SopA-containing complexes were immunoprecipitated using anti-green fluorescent protein (GFP) beads, subjected to tryptic in-gel digest and extracted peptides were analysed by liquid chromatography-coupled MS/MS on an Orbitrap mass spectrometer (Fig. 1a and Supplementary Fig. 1b). We reproducibly identified two TRIM E3 ligases, TRIM56 and TRIM65, as the most significant SopA-enriched hits (Fig. 1b, Supplementary Fig. 1c and Supplementary Data 1). Subsequent immunoblotting experiments confirmed these MS data and revealed that endogenous TRIM56 and TRIM65 indeed specifically co-precipitated with transiently expressed GFP-SopA, but not with its enterohemorrhagic Escherichia coli homologue NleL (Fig. 1c). To additionally identify SopA interactors in the course of infection, we isolated SopA from cells infected with S. Typhimurium strains expressing tagged SopA–HA or empty vector controls. Consistent with results from heterologous expression experiments, we recovered TRIM56 and TRIM65 as major interacting proteins of bacterially secreted SopA in Salmonella-infected cells (Fig. 1d,e and Supplementary Data 2).

Given the reported ability of TRIM proteins to form hetero-oligomers[27], we tested whether TRIM56 and TRIM65 interact with each other. We did not detect complex formation between these two TRIM proteins after transient expression of tagged TRIM versions (Supplementary Fig. 1e,f). To further characterize the interaction mode between TRIM56/TRIM65 and SopA, we expressed different truncation constructs of both TRIM proteins individually and tested them for the ability to co-immunoprecipitate SopA (Fig. 2a and Supplementary Fig. 1d). Whereas ablation of both coiled-coil and substrate-binding domains had no effect, deletion of the RING domain in case of both TRIM56 and TRIM65 completely abolished SopA interaction (Fig. 2c,d). Analogously, we expressed SopA truncations to determine the requirements for TRIM56/65 interaction (Fig. 2b). Interestingly, the capacity of SopA to associate with both TRIM56/65 mapped to the N-terminal β-helix domain (Fig. 2e). Consistently, pull-down experiments using recombinant TRIM56 and SopA proteins revealed that SopA directly interacts with the RING of TRIM56 or TRIM65 via its β-helix domain (Fig. 2f and Supplementary Fig. 1g).

### Crystal structure of the SopA–TRIM56 complex.
To understand the molecular basis of the interaction between SopA and TRIMs, we set out to determine the crystal structure of the SopA–TRIM56 complex. Initial attempts to co-purify bacterially expressed SopA (163–782) and TRIM56 RING domain (1–94) did not yield stoichiometric complex in size-exclusion chromatography (Supplementary Fig. 2a), due to the low affinity binding between these molecules with a dissociation constant of ~9 μM (Fig. 2g). To circumvent this problem, we fused the minimal binding regions of TRIM56 (1–94) and SopA (163–425) using a

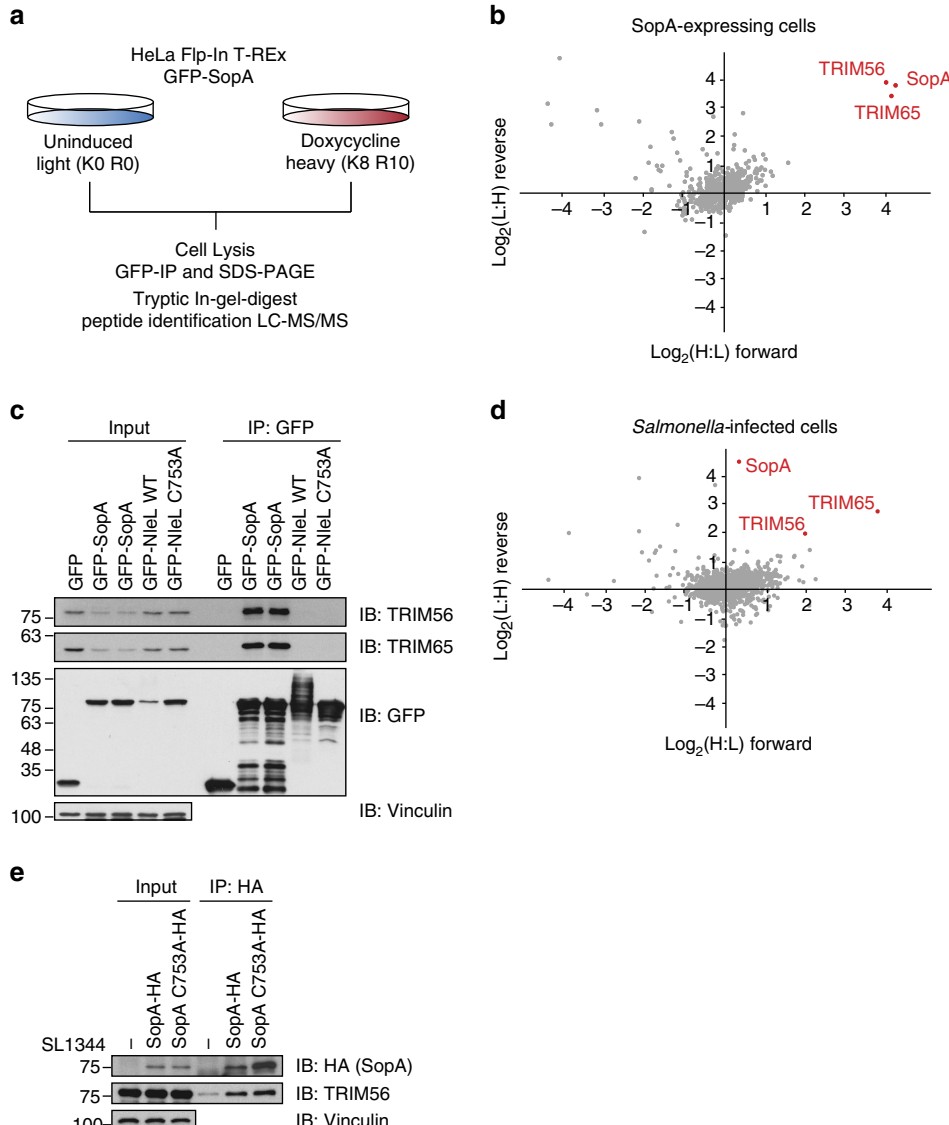

**Figure 1 | Identification of TRIM56 and TRIM65 as SopA-interacting proteins.** (**a**) Workflow for SILAC-coupled SopA interactome analysis from inducible HeLa Flp-In T-REx GFP-SopA-expressing cells. (**b**) SopA interacts with TRIM56 and TRIM65. Scatter plot of forward and reverse SILAC SopA interactome. Proteins situated in the upper left quadrant include contaminants. (**c**) Endogenous TRIM56 and TRIM65 specifically interact with SopA. Lysates from HEK293T cells expressing GFP, GFP-SopA or GFP-NleL constructs were subjected to anti-GFP IP, followed by SDS–PAGE and immunoblotting. (**d**) Bacterially translocated SopA interacts with TRIM56/65. Scatter plot of forward and reverse SILAC interactome experiments from *Salmonella*-infected HeLa cells. Proteins situated in the upper left quadrant include contaminants. (**e**) Endogenous TRIM56 interacts with bacterially secreted SopA during infection. Lysates from HeLa cells infected with SL1344 WT, SopA–HA or catalytic-dead SopA C753A-HA-expressing strains were subjected to anti-HA IP, followed by SDS–PAGE and immunoblotting.

flexible linker (Supplementary Fig. 2b). Such end-to-end fusion of two different proteins with short flexible linkers is a commonly employed method in crystallization of low-affinity protein complexes[28]. This fusion construct was purified and crystallized in the space group P $3_1$ 1 2 and the optimized crystals diffracted to ~2.9 Å resolution (Supplementary Fig. 2c). The structure was determined by molecular replacement and refined until convergence (Fig. 3a and Table 1).

The asymmetric unit contained one copy of the SopA–TRIM56 dimer. The structure reveals that the first $Zn^{2+}$-binding loop in the TRIM56 RING domain is packed in a cleft at the interface of the β-helix and the N-lobe domains of SopA (Fig. 3b). The structure of SopA in complex with TRIM56 overlays well with the SopA apo structure with a mean root mean squared deviation (r.m.s.d.) of 0.9 Å over all the C-α atoms[9]. Both the linker residues

between TRIM56 and SopA, as well as the 17 N-terminal residues of TRIM56 were disordered and could not be observed in the electron density. Comparison of the TRIM56 RING domain structure with all the structures in PDB using DALI revealed that it is most similar to the RING domain of RNF146 with a mean r.m.s.d. of 2.5 Å over all the C-α atoms[29]. Moreover, the RING domains of TRAF6, RING1B, TRIM32 and RNF4 also closely resemble the TRIM56 RING domain with mean r.m.s.d. of 2.6, 1.8, 2.1 and 1.9 Å over >90% of C-α atoms, respectively. The interface of SopA and TRIM56 contains a mix of hydrophobic/hydrophilic interactions with a buried surface area of 750 Å$^2$ (Fig. 3b). In TRIM56, residues Leu25 and Glu26 contribute majorly towards interaction with SopA. Although Glu26 makes polar contacts with Arg296, His297 and Lys298 of SopA, Leu25 inserts into a hydrophobic pocket of SopA involving Phe345 and Pro334.

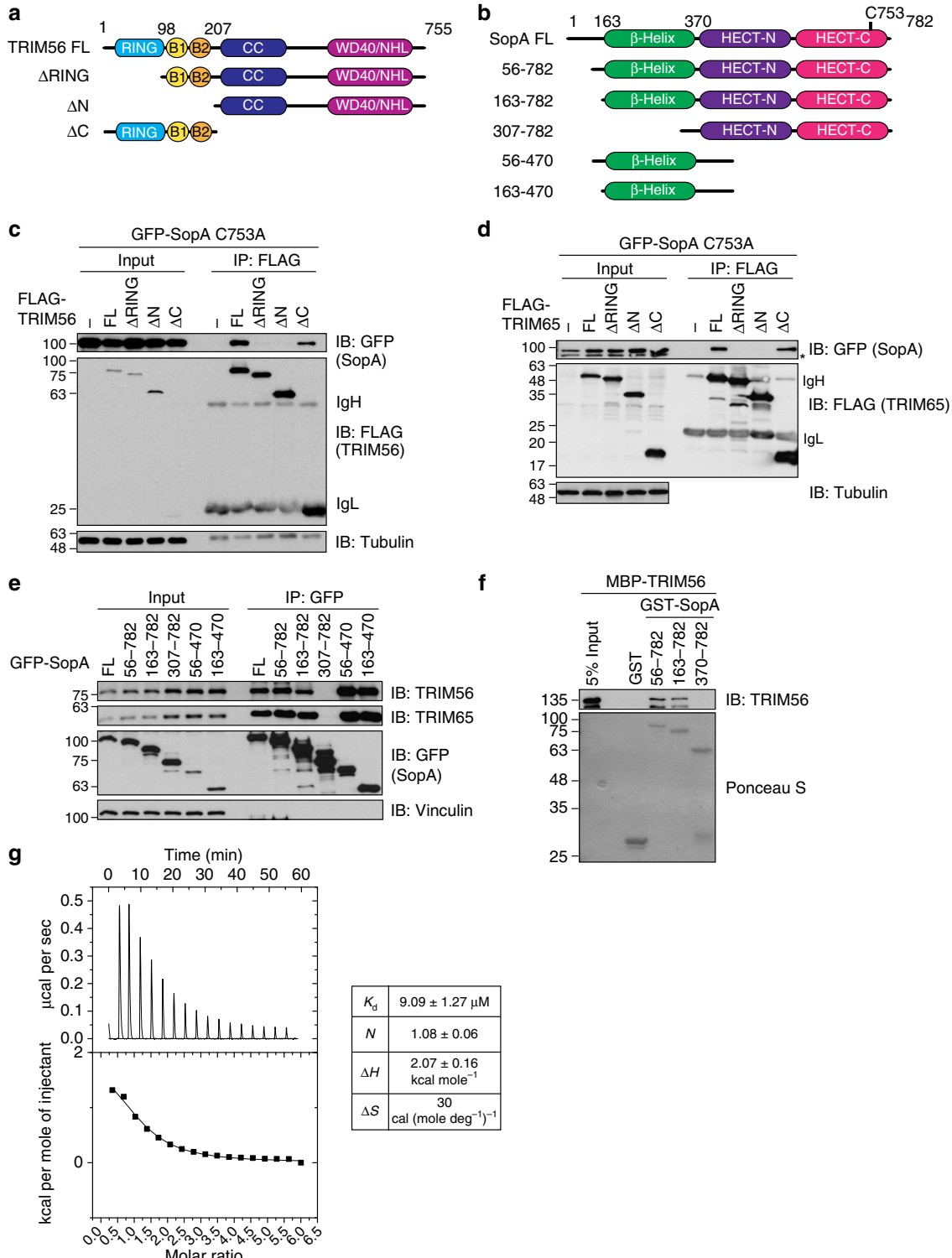

**Figure 2 | TRIM56 and TRIM65 RING domains directly interact with the β-helix region of SopA.** (**a**,**b**) Domain organization of TRIM56 (**a**) and SopA (**b**), and corresponding truncation constructs. B1 and B2, B-box type zinc fingers; CC, coiled-coil domain; HECT-N, N-lobe; HECT-C, C-lobe. (**c**,**d**) TRIM56/TRIM65 RING domain is required for SopA interaction. Lysates from HEK293T cells co-expressing GFP-SopA C753A and indicated FLAG-TRIM56 (**c**) or FLAG-TRIM65 (**d**) constructs were subjected to anti-FLAG IP, followed by SDS–PAGE and immunoblotting. (**e**) Intact SopA β-helix is required for TRIM56/TRIM65 interaction. Lysates from HEK293T cells expressing indicated GFP-SopA constructs were subjected to anti-GFP IP, followed by SDS–PAGE and immunoblotting. (**f**) SopA–TRIM56/65 interaction is direct. Recombinant MBP-TRIM56 was incubated with GST or different GST-SopA proteins and subjected to glutathione sepharose pull-down followed by SDS–PAGE and immunoblotting. (**g**) Isothermal titration calorimetry (ITC) measurement of SopA–TRIM56 RING interaction. 1,000 μM TRIM56 (1–94) was injected gradually into 20 μM SopA (163–782) present in the sample cell. Raw ITC data were plotted and analysed using Origin 7 software. The table shows values obtained for various parameters of the SopA–TRIM56 interaction. Δ*H*, heat change; *K*$_d$, dissociation constant; *N*, occupancy; Δ*S*, entropy change.

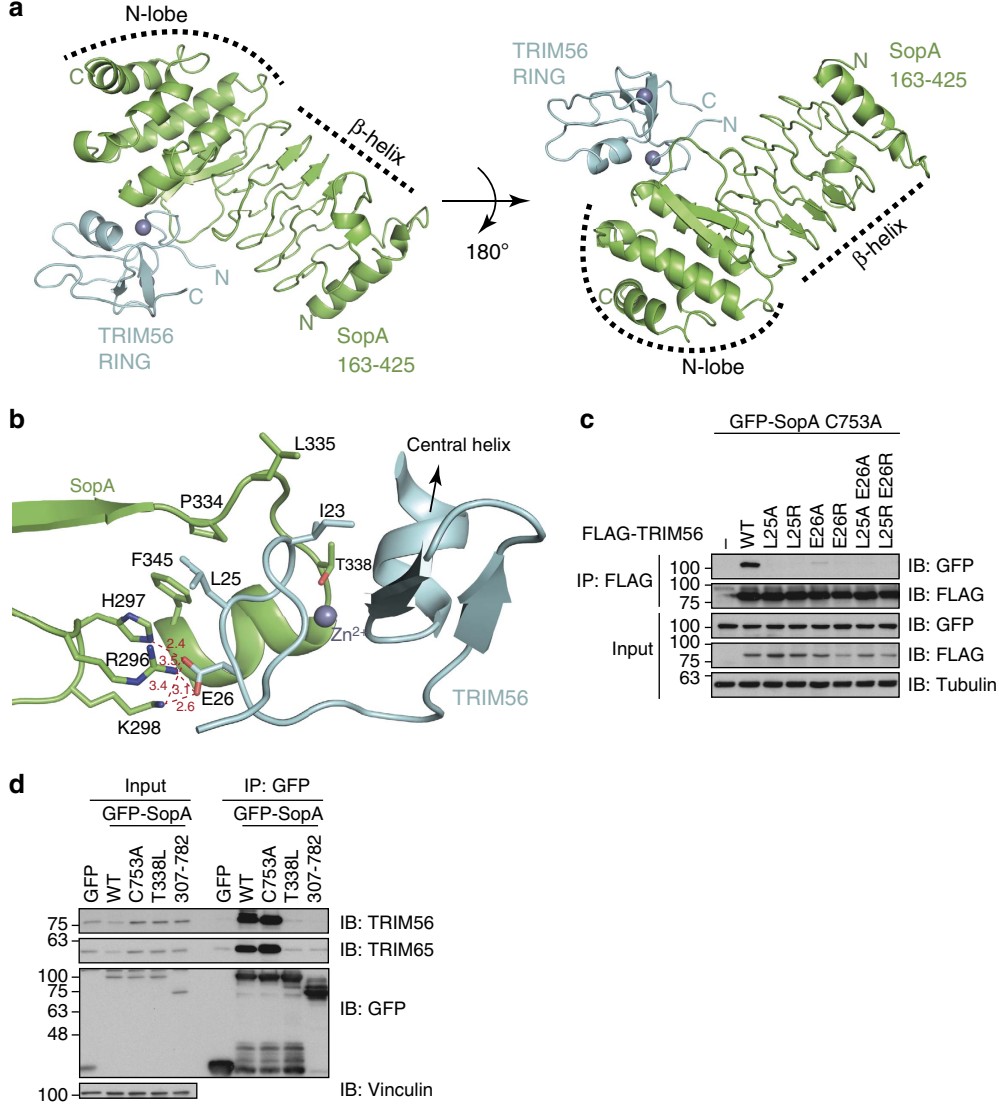

**Figure 3 | Structure of SopA β-helix in complex with TRIM56 RING domain.** (**a**) Crystal structure of SopA–TRIM56 complex shown in cartoon representation. SopA is shown in green and TRIM56 in cyan, two orientations rotated by x:180° are shown. (**b**) Enlarged view of the interface of SopA and TRIM56 showing the important residues involved in the interaction. Right conformation depicted in **a** was rotated by y: 140°, x: 145° and z: 35° and zoomed in; residues 18–48 in TRIM56 RING and 295–299, as well as 330–345 in SopA are shown. Hydrogen bonds with corresponding distances are depicted in deep red. (**c**) TRIM56 RING mutations abolish SopA interaction. Lysates from cells co-expressing GFP-SopA C753A and indicated FLAG-TRIM56 RING domain mutants were subjected to anti-FLAG IP, followed by SDS–PAGE and immunoblotting. (**d**) SopA T338L mutation abolishes SopA–TRIM56 and SopA–TRIM65 interaction. Lysates from cells expressing indicated GFP-SopA β-helix point mutants were subjected to anti-GFP IP, followed by SDS–PAGE and immunoblotting.

The central α-helix present in TRIM56 also makes several hydrophobic contacts and packs tightly against SopA (Fig. 3b).

Structure-guided point mutations in the first $Zn^{2+}$-binding loop of TRIM56 RING confirmed the requirement of Leu25 and Glu26 for SopA–TRIM complex formation. The corresponding TRIM56 single and double mutants were unable to co-precipitate SopA upon transient expression, while retaining functional RING E3 activity (Fig. 3c and Supplementary Fig. 2d). Thr338 of SopA is in close proximity to the central α-helix in TRIM56 (Fig. 3b) and we predicted that mutating it to Leucine would create steric clashes and impair the interaction. Indeed, a T338L point mutation in SopA completely abolished interaction with endogenous TRIM56 (Fig. 3d). Interestingly, SopA T338L mutation also prevented binding of SopA to TRIM65, indicating that SopA recognizes both TRIM56 and TRIM65 in a similar manner.

**Structural basis of SopA specificity for TRIM56 and TRIM65.** Sequence comparison of SopA and its *E. coli* homologue NleL revealed that NleL lacks all the residues involved in SopA–TRIM56 binding (Supplementary Fig. 2e). This readily explained why NleL does not target TRIM proteins (Fig. 1c). On the other hand, multiple sequence alignment of several TRIM proteins and other closely related RING domain-containing proteins revealed that residues Leu25 and Glu26, which are essential for TRIM56-SopA interaction, show a conservation of 75% and 50%, respectively (Fig. 4a). Despite this conservation, no other detected TRIM protein was found significantly enriched in our SopA interactome studies (Supplementary Data 1,2), indicating that the primary sequence of TRIM proteins may not determine SopA specificity. Other factors influencing SopA selectivity may include subcellular targeting and expression levels of various TRIMs.

**Table 1 | Data collection and refinement statistics.**

|  | SopA–Trim56 complex |
|---|---|
| Wavelength (Å) | 0.99987 |
| Resolution range (Å) | 43.48-2.849 (2.95-2.85) |
| Space group | P 31 1 2 |
|  |  |
| *Unit cell* |  |
| a, b, c (Å) | 71.01, 71.01, 122.94 |
| α, β, γ (°) | 90, 90, 120 |
| Unique reflections | 8,375 (833) |
| Multiplicity | 20.0 (19.0) |
| Completeness (%) | 98.37 (84.05) |
| Mean I$\sigma$(I) | 31.15 (3.64) |
| Wilson B-factor (Å$^2$) | 77.87 |
| R-merge | 0.074 (0.834) |
| CC1/2 | 1 (0.985) |
| R-work | 0.2234 (0.3022) |
| R-free | 0.2780 (0.3269) |
| Number of atoms | 2395 |
| Macromolecules | 2392 |
| Ligands | 2 |
| Water | 1 |
| Protein residues | 325 |
| Root mean square (bonds; Å) | 0.004 |
| Root mean square (angles; °) | 0.97 |
| Ramachandran favoured (%) | 92 |
| Ramachandran outliers (%) | 0.94 |
| Clashscore | 6.51 |
| Average B-factor (Å$^2$) | 86.30 |
| Macromolecules | 86.30 |
| Ligands | 88.50 |
| Solvent | 46.80 |

Statistics for the highest-resolution shell are shown in parentheses.

Here we investigated whether the crystal structure of the SopA–TRIM56 complex explains the specificity of SopA. We superimposed the TRIM56 RING domain with RINGs of various TRIM proteins (only TRIM32 and TRIM39 are shown as representatives), as well as RNF4 and TRAF6 (Fig. 4b–e). Although, the first $Zn^{2+}$-binding loop of all the aligned RING domains fits into the cavity between the β-helix domain and the N-lobe of SopA without sterical clashes, the central α-helix of various RING domains appears to be clashing with SopA and rendering the binding unfavourable. Hence, positioning of the central α-helix of RING domains relative to the first $Zn^{2+}$-binding loop is one of the crucial factors defining the specificity of SopA. Consistent with these data, SopA was not able to bind and co-precipitate TRIM32, TRIM39 and RNF4 (Fig. 4f). In essence, a combination of sequence and structural features determine SopA-binding specificity towards the RING domains of TRIM56 and TRIM65.

**SopA inhibits TRIM56 ligase activity.** To gain more insights from the structure of SopA–TRIM56 complex, we superimposed TRIM56 RING with the RNF4 RING bound to UbcH5a (PDB: 4AP4)[30] (Fig. 5a). Surprisingly, this revealed that both SopA-binding and potential E2-binding surfaces on TRIM56 are overlapping, suggesting that SopA may hinder the interaction of E2 and TRIM56. In other words, SopA may negatively regulate the Ub ligase activity of TRIM56 by masking its E2-binding surface. To address this hypothesis, we first tested if the bacterially purified TRIM56 constructs harbour E3 ligase activity *in vitro*. Various TRIM56 constructs were incubated in the presence of three different E2s: UbcH5a, UbcH5b and UbcH7 (Fig. 5b). All TRIM56 constructs including the minimal RING

domain used for crystallization of the SopA–TRIM56 complex were active and robustly synthesized free Ub chains in the presence of UbcH5a and UbcH5b, but not in the presence of UbcH7. We then examined how increasing amounts of SopA affect the Ub ligase activity of full-length TRIM56 (Fig. 5c). SopA (163–425) lacking its E2-binding region[14] was used in this assay to negate the possibility of SopA-mediated E2 recruitment. Although a previous report proposed SopA to activate TRIM56 and TRIM65 activity[15], we observed that SopA (163–425) inhibited the ligase activity of TRIM56 as seen by the decrease in free Ub chains and especially di-Ub formation (Fig. 5c).

**SopA mediates degradative ubiquitination of TRIM56.** To identify SopA substrates, we compared the diGly-modified proteome of cells infected with wild-type *Salmonella* (SL1344 WT) or a *sopA* deletion strain (*ΔsopA*) (Fig. 6a and Supplementary Fig. 3a). Two replicate SILAC diGly proteomics experiments resulted in the quantification of ~9,000 diGly sites in ~4,500 proteins (Supplementary Fig. 3b). Importantly, we identified multiple ubiquitination sites in TRIM56 (K87, K270 and K377) and TRIM65 (K206) as SopA-regulated events with the highest SILAC ratios of all quantified peptides (Fig. 6b, Supplementary Fig. 3c and Supplementary Data 3). To elaborate these findings, we purified ubiquitinated proteins from cells, which were infected with SL1344 WT, a non-invasive mutant defective in effector secretion (*ΔSPI1*) or with various *sopA* deletion (*ΔsopA*) and complemented strains (*ΔsopA + sopA*, *ΔsopA + sopA C753A*) in the presence or absence of the proteasome inhibitor MG132 using tandem Ub-binding entities (TUBEs; Fig. 6c)[31]. Indeed, we observed the appearance of ubiquitinated high molecular weight species of endogenous TRIM56 upon *Salmonella* infection (Fig. 6c). Infection of cells with complemented *sopA* deletion strains, expressing higher amounts of SopA relative to SL1344 WT *Salmonella* (Supplementary Fig. 3g), resulted in increased modification of TRIM56. Interestingly, ubiquitination of TRIM56 increased upon proteasome inhibition and was completely dependent on the presence of catalytically competent SopA. Moreover, infection of cells with *Salmonella* expressing the binding-deficient SopA T338L mutant did not result in TRIM56 modification (Fig. 6d). In agreement with the results obtained from infected cells, we were able to reconstitute SopA-mediated ubiquitination of TRIM56 *in vitro* using recombinant proteins. Importantly, to monitor SopA catalytic activity and to exclude TRIM-mediated chain formation, we used the E2 enzyme UbcH7 in these experiments (Fig. 5b). Both the isolated RING domain and full-length TRIM56 were robustly ubiquitinated by SopA *in vitro* (Fig. 6e and Supplementary Fig. 3d), whereas TRIM56 RING mutants unable to interact with SopA were not modified (Supplementary Fig. 3e). As conjugation of substrates with different types of polyubiquitin chains can have distinct functional outcomes[32], we decided to examine the Ub linkage preference of SopA as well as TRIM56. For this purpose, we performed *in vitro* ubiquitination reactions using Ub mutants harbouring single lysine residues. Although TRIM56 catalytic activity showed a bias towards K11- and K63-linked polyubiquitin (Supplementary Fig. 3f), SopA preferentially modified TRIM56 with K48- and K11-linked Ub chains implicated in proteasomal targeting (Fig. 6f).

**Infection induces proteasomal turnover of TRIM56/TRIM65.** Given the MG132-sensitive nature of SopA-mediated TRIM56 ubiquitination and its preference for the synthesis of mostly degradative K48- and K11-chain types, we monitored the abundance of TRIM56 and TRIM65 proteins after infection of cells with *S.* Typhimurium WT, *sopA*-deficient or *sopA*-complemented

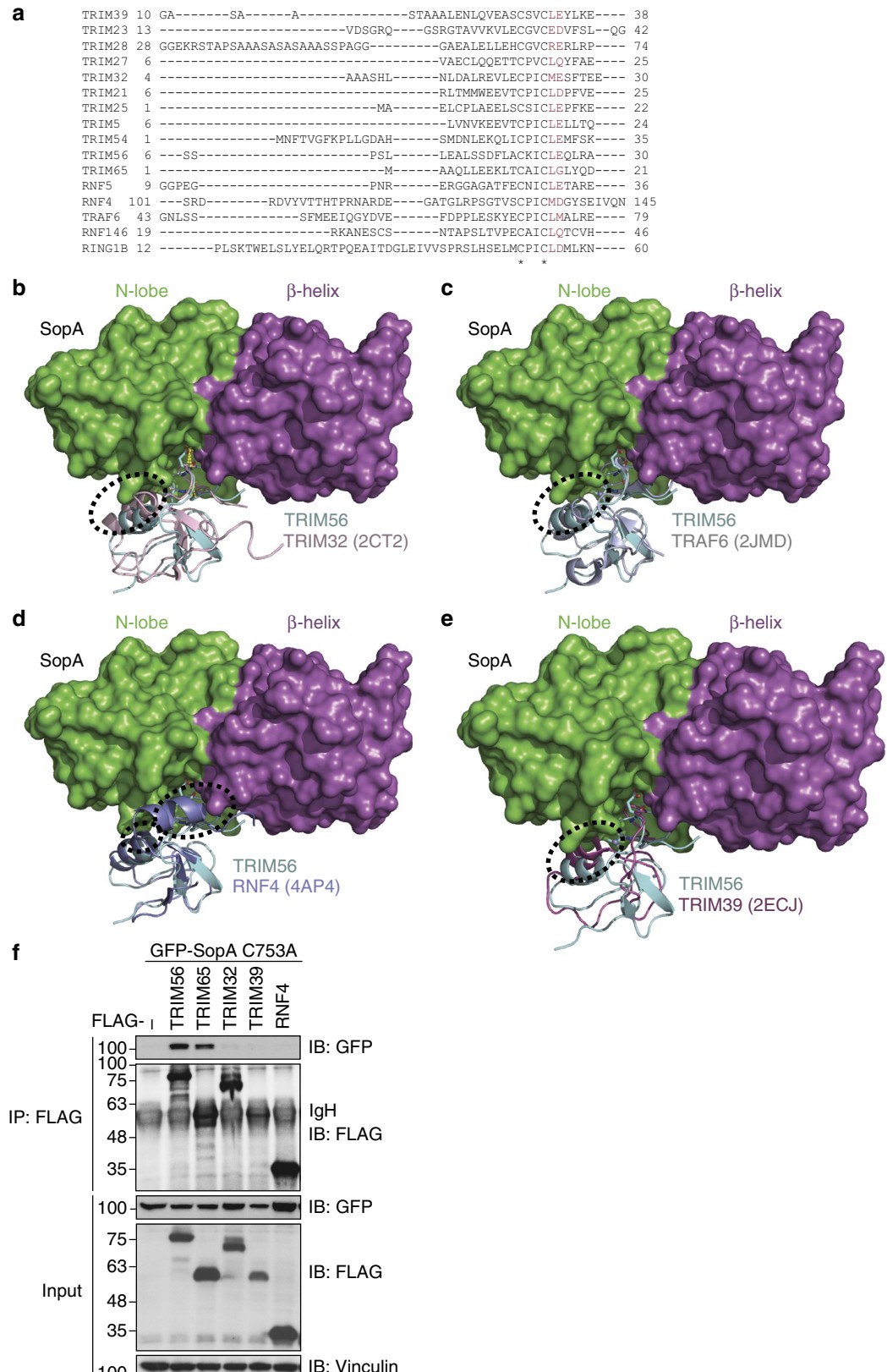

**Figure 4 | Structural features of TRIMs define specificity of SopA towards TRIM56.** (**a**) Multiple sequence alignment of the first Zn$^{2+}$-binding loop region from various TRIMs and the closely related RING domain-containing proteins reveal Glu25 and Leu26 (residues in TRIM56 that are involved in binding to SopA) are fairly conserved (coloured in red). Zn$^{2+}$-coordinating cysteines are indicated using an asterisk. (**b,c,d,e**) TRIM56 RING domain was aligned with various closely related RINGs showing the clashes (circled) of SopA with the central α-helix of various RING domains (**b**, TRIM32; **c**, TRAF6; **d**, RNF4; **e**, TRIM39). PDB codes of the structures used in the alignment are indicated in brackets: (2JMD)[47], (4AP4)[30]. (**f**) SopA does not bind TRIM32, TRIM39 and RNF4. Lysates from cells co-expressing GFP-SopA C753A and indicated FLAG-RING E3 ligase constructs were subjected to anti-FLAG IP, followed by SDS–PAGE and immunoblotting.

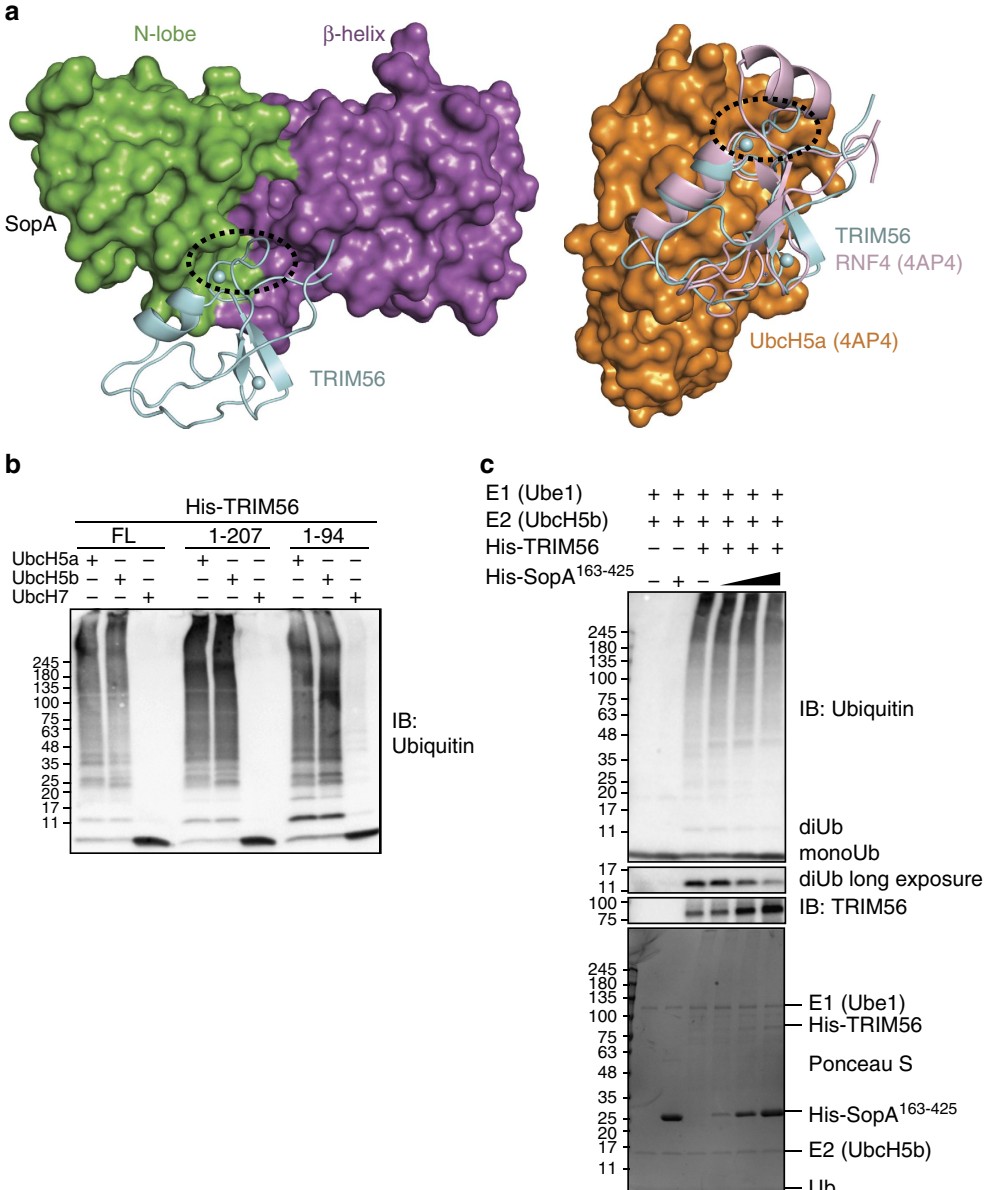

**Figure 5 | SopA inhibits TRIM56-mediated ubiquitin chain formation.** (**a**) SopA occupies the potential E2-binding site on TRIM56. Left, crystal structure of SopA–TRIM56 with SopA shown in surface representation. β-Helix and part of the N-lobe are coloured differently as indicated. Right, TRIM56 RING domain is superimposed with RNF4 RING domain bound to UbcH5a (PDB code: 4AP4)[30]. First $Zn^{2+}$-binding loop of TRIM56 RING domain is circled to highlight overlapping binding interfaces of SopA and E2. (**b**) Ubiquitin ligase activity of various TRIM56 constructs in the presence of UbcH5a, UbcH5b and UbcH7. (**c**) Effect of SopA-binding on the ligase activity of TRIM56 full-length protein. SopA (163–425) was added at $1\times$, $5\times$ and $10\times$ molar excess of E2. Inhibition of TRIM56 activity was monitored by free ubiquitin chain formation. Apparent increase in TRIM56 full-length levels with increasing SopA is due to a decrease in auto-ubiquitination of TRIM56.

strains. These experiments revealed that protein levels of both TRIMs decreased in a SopA ligase activity-dependent manner (Fig. 7a). To address whether SopA-mediated reduction of TRIM56 and TRIM65 arises due to diminished protein translation or due to destabilization of TRIM56 and TRIM65 proteins, we performed cycloheximide chase experiments. Infection of cycloheximide-treated cells induced robust SopA-driven degradation of TRIM56 and TRIM65 (Fig. 7b). We were able to recapitulate these findings in doxycycline-inducible SopA-expressing cells, in which SopA induction triggered TRIM56 and TRIM65 degradation. Additional treatment of these cells with MG132 restored TRIM protein levels and resulted in the appearance of high molecular weight ubiquitinated species (Fig. 7c). Interestingly, a previous study suggested SopA targeting

of TRIM56 and TRIM65 to be non-degradative[15]. Here we provide multiple lines of evidence, which strongly support the notion that SopA-mediated ubiquitination inhibits and triggers the proteasomal degradation of TRIM56 and TRIM65 during *Salmonella* infection (Fig. 8).

## Discussion
By using an unbiased multi-layered proteomics approach in *Salmonella*-infected cells, we recovered two host TRIM RING ligases, TRIM56 and TRIM65, as interactors and substrates of the *Salmonella* HECT-like ligase SopA. Using co-immunoprecipitation of various constructs of SopA and TRIMs, we showed that the N-terminal RING domains of TRIM proteins are both

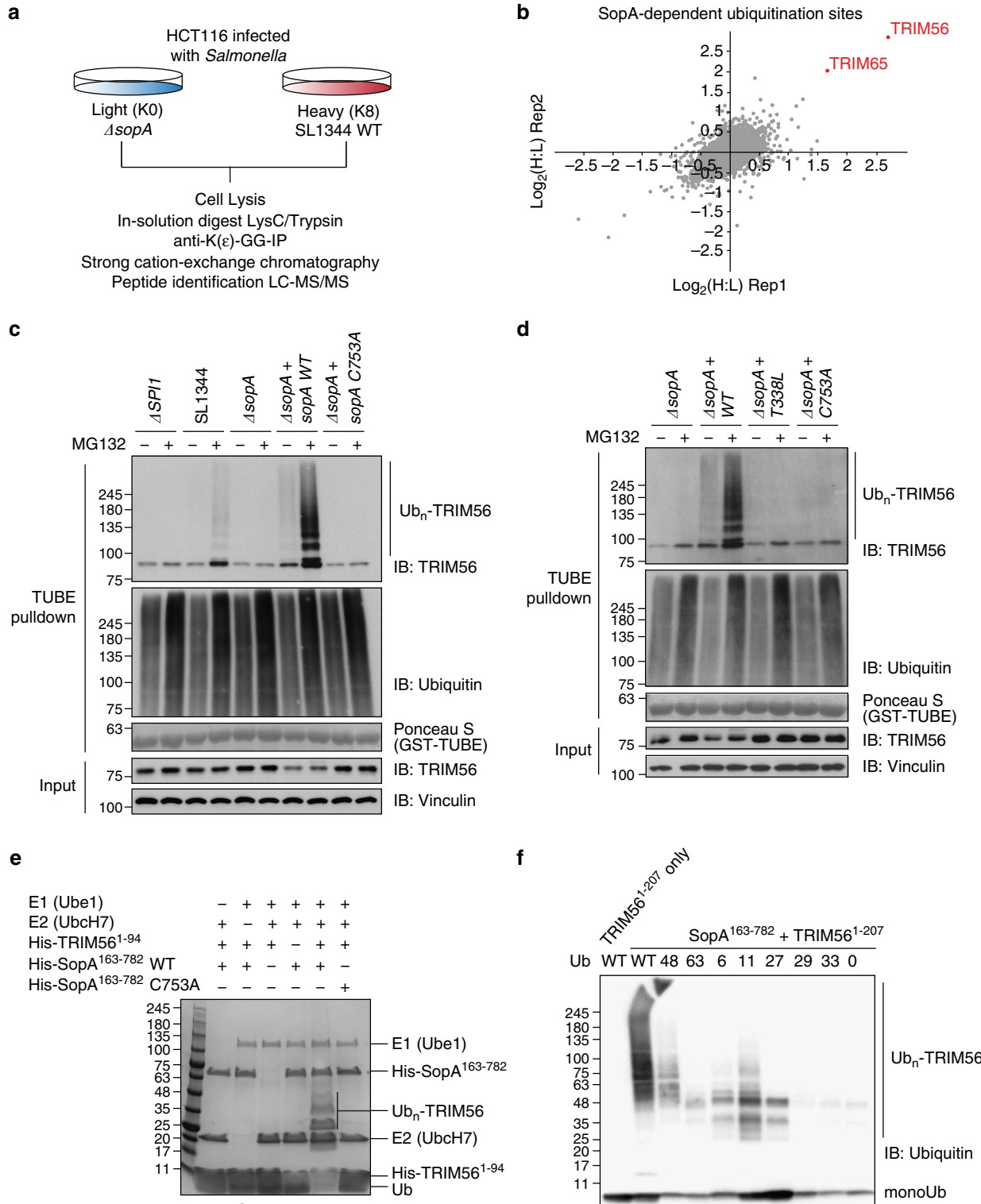

**Figure 6 | TRIM56 and TRIM65 are substrates of degradative SopA ubiquitination.** (**a**) Workflow for SopA-dependent ubiquitinome analysis using SILAC diGly proteomics of *Salmonella* SL1344 WT and *ΔsopA*-infected HCT116 cells 30 min post infection. (**b**) SopA ubiquitinates TRIM56 and TRIM65. Scatter plot of replicate SopA ubiquitinome experiments. (**c,d**) SopA-mediated ubiquitination of TRIM56 on infection is MG132 sensitive and depends on SopA catalytic activity (**c**) and on SopA–TRIM56 binding (**d**). Lysates from HeLa cells infected with *Salmonella* SL1344 WT or indicated mutant strains in the absence or presence of 20 μM proteasome inhibitor MG132 were subjected to TUBE pulldown followed by SDS–PAGE and immunoblotting. (**e**) SopA ubiquitinates TRIM56 RING domain *in vitro*. Purified SopA WT and catalytic mutant (C753A) were incubated with E1, UbcH7, ubiquitin, ATP and TRIM56 RING domain. TRIM56 ubiquitination is seen with WT SopA but not in the presence of catalytic-dead SopA. (**f**) Testing SopA ubiquitin chain specificity. WT SopA was incubated with TRIM56 (1–207) and WT ubiquitin or various ubiquitin mutants that contain only a single surface lysine. E1, UbcH7, ubiquitin and ATP were added to all the reactions. TRIM56 alone is not active under these conditions (lane 1).

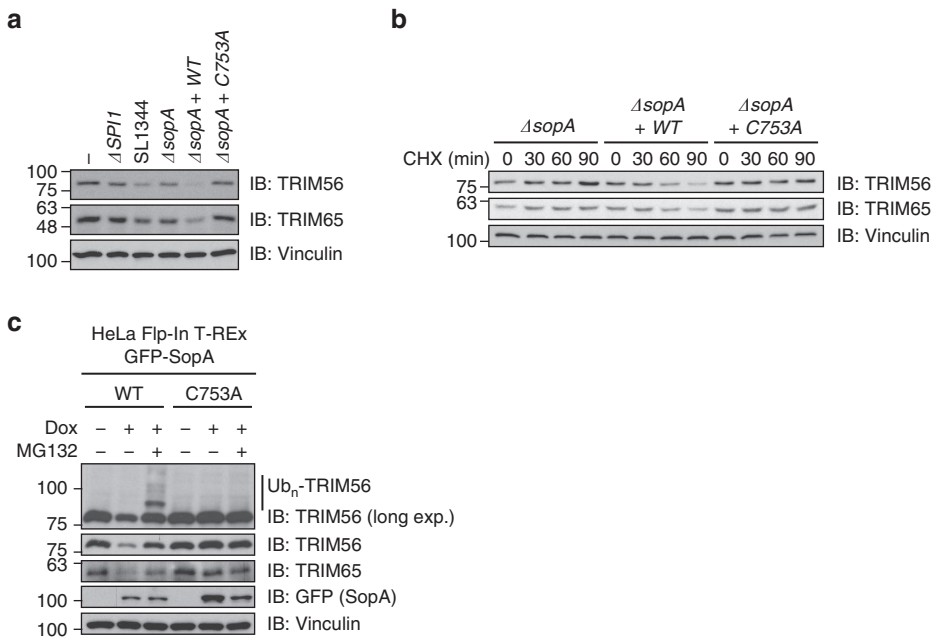

**Figure 7 | *Salmonella* infection induces proteasomal degradation of TRIM56 and TRIM65.** (**a**) TRIM56 and TRIM65 protein levels decrease in SopA activity-dependent manner. Lysates from HCT116 cells infected with *Salmonella* SL1344 WT, ΔSPI1 or indicated *sopA* mutant strains were subjected to SDS–PAGE and immunoblotting. (**b**) SopA-mediated degradation of TRIM56 and TRIM65 on infection. Lysates from HCT116 cells treated with 100 µg ml$^{-1}$ cycloheximide and infected with the indicated *sopA* mutant *Salmonella* strains for indicated time points were subjected to SDS–PAGE and immunoblotting. (**c**) SopA-mediated degradation of TRIM56 and TRIM65 on heterologous expression. Lysates from inducible HeLa Flp-In T-REx GFP-SopA cells left untreated or treated with indicated combinations of 1 µg ml$^{-1}$ doxycycline and 20 µM MG132 were subjected to SDS–PAGE and immunoblotting.

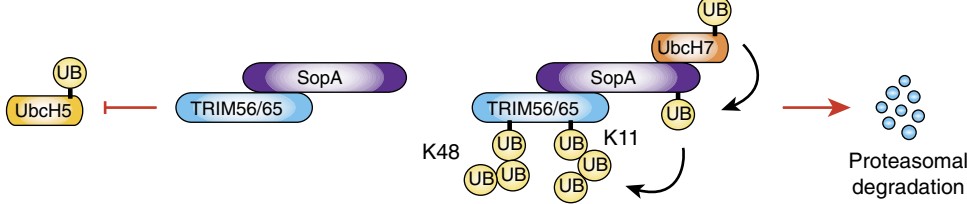

**Figure 8 | Model for dual SopA-mediated targeting of TRIM56 and TRIM65.** Binding of SopA to the RING domains of TRIM56 and TRIM65 impedes TRIM E3 Ub ligase activity by occluding the RING-E2 interaction surface (left panel). The SopA HECT ligase activity deposits degradative ubiquitin chains linked via K48 and K11 on TRIM56 and TRIM65, resulting in their proteasomal degradation during *Salmonella* infection (right panel).

necessary and sufficient for binding to the N terminus of SopA. It is interesting to note that SopA specifically interacts with the RING domains of TRIM56 and TRIM65 amongst a large number of expressed RING domain-containing human proteins[33] (Supplementary Fig. 4a,b). To understand the basis for this remarkable specificity, we determined the crystal structure of SopA (residues 163–425) and TRIM56 (residues 1–94) in complex. The structure revealed the interface of SopA and TRIM56 in atomic detail and also provided an explanation for the specificity of SopA towards TRIM56 and TRIM65. The placement of the central α-helix in various RING domains in combination with the defined sequence features of the first Zn$^{2+}$-binding loop appears to determine the specificity for SopA binding. Comparison of our SopA–TRIM56 complex structure with a previously determined SopA structure carrying the C-lobe[9] uncovers that the catalytic cysteine in the C-lobe is positioned in close proximity to the C terminus of the TRIM56 RING (Supplementary Fig. 4c). Strikingly, we found that the ubiquitination site at Lys87 in the RING domain of TRIM56 is one of the most upregulated modification events in cells infected with SopA-containing *Salmonella* (Fig. 6b and Supplementary Data 3), thus supporting the juxtaposition of these domains in

our model (Supplementary Fig. 4c). However, Lys87 is disordered in the crystal structure of SopA–TRIM56, indicating that Ub conjugation to the HECT-active site may be necessary to stabilize the substrate lysine. Structural investigation of TRIM56 in complex with Ub-conjugated SopA will provide more insights into the SopA mechanism of action and HECT catalytic mechanism in general.

Interestingly, a recent study demonstrated the requirement of RING dimerization for the E3 ligase activity of TRIM25 and TRIM32 (ref. 26). We did not observe a dimer of TRIM56 RING in our crystal structure, raising the possibility that SopA binding may interfere with TRIM56 RING dimerization. To address this, we aligned one RING domain of the TRIM32 dimer with TRIM56 RING in the SopA–TRIM56 complex structure (Supplementary Fig. 4d). In this setup, we observed only minor clashes between the TRIM32 dimer and SopA, indicating that SopA interaction may be compatible with TRIM56 dimerization in solution. Accordingly, we predict the potential TRIM56 dimer to bind two molecules of SopA under saturating conditions. The absence of dimerization of the TRIM56 RING in our structure might therefore be a result of the C-terminal fusion of SopA to the RING domain for the purpose of crystallization.

Further analysis of the SopA–TRIM56 structure indicated that SopA binding to the RING domain obstructs the E2-binding site of TRIM56. This led us to hypothesize that SopA interferes with TRIM ligase activity. Titration experiments indeed provided support for SopA interaction-mediated inhibition of TRIM56 activity. It should be noted that in our assays SopA (residues 163–425) was added at $1 \times$, $5 \times$ and $10 \times$ molar excess compared with E2 and the effective inhibition was only achieved with the last two conditions. It remains to be seen whether the inhibitory action of SopA on TRIM56 ligase activity is promoted by localized interactions in the context of full size proteins during infection. This interaction mode of SopA is however an effective way for proteins to ubiquitinate E3 ligases in general without getting ubiquitinated themselves. This is achieved, because binding of SopA and E2 to the TRIM56 RING domain is mutually exclusive. Moreover, acute inhibition of TRIM56 and TRIM65 by binding of SopA functions in conjunction with SopA-mediated degradative ubiquitination of TRIM56/65 to downregulate activity and abundance of TRIM proteins in vivo. Taken together, our results provide evidence for an inhibitory mechanism underlying targeting of TRIM56/65 via SopA.

It is interesting to note that a recent study proposed a different model of action for SopA-mediated TRIM56/65 ubiquitination[15]. SopA-catalysed ubiquitination of TRIM56 and TRIM65 was suggested to be non-degradative and to promote TRIM56 and TRIM65 activity towards their respective targets RIG-I and MDA5, resulting in an increased transcription of type I interferon. Here we provide multiple lines of evidence in support of SopA-mediating degradation of TRIM56 and TRIM65. We observed that SopA-dependent ubiquitination of TRIM56 is robustly increased after treatment of infected cells with proteasome inhibitor. Moreover, both proteomics and biochemical data revealed that TRIM56 and TRIM65 are degraded in the course of infection and heterologous SopA expression with protein levels being restored by proteasome inhibition. In vitro ubiquitination experiments further indicated that SopA preferentially modifies TRIM56 with K48- and K11-linked polyubiquitin, two chain types known to confer proteasomal turnover[32]. Importantly, we did not detect stimulation of TRIM56 or TRIM65-mediated type I interferon expression by SopA, but observed that SopA activity confers decreased interferon-β transcription instead. Although it can not be excluded that the relative expression levels and E3 ligase activities of secreted SopA and host TRIM56 and TRIM65 in vivo determine the outcome of this host–pathogen targeting event and account for the discrepancy in presented findings and previous data, our observations in cultured epithelial cells indicate a progressive decay of endogenous TRIM56 and TRIM65 protein levels upon standard Salmonella infection conditions.

In general, structural information of proteins has proven instrumental to the identification and design of lead compounds for the inhibition of enzymatic activities and interaction interfaces[34]. Given that Salmonella pathogenesis in vivo requires the induction of mucosal inflammation to establish a proliferative niche, we envision that the structure of the SopA–TRIM56 complex presented here could aid approaches for the chemical modulation of this host–pathogen interaction.

## Methods

**Cell lines.** HeLa, HCT116 and HEK293T cells were obtained from ATCC and cultured in DMEM medium (Thermo Scientific) with 10% fetal bovine serum (Thermo Scientific), 0.2 mM L-glutamine (Thermo Scientific) and penicillin/streptomycin (Sigma Aldrich) at 37 °C and 5% $CO_2$. HeLa FRT/TO cells for the generation of stable cell lines using the Flp-In T-REx System (Thermo Scientific) were generously provided by S. Taylor (Faculty of Life Sciences, University of Manchester, UK) and maintained in medium containing 15 μg ml$^{-1}$ Blasticidin. Stable HeLa Flp-In T-REx GFP-SopA cells were generated by transfection of HeLa

FRT/TO cells with Flp-recombinase expression vector pOG44 and pcDNA5 FRT/TO GFP-SopA constructs in a 9:1 ratio. Thirty-six hours post transfection, cells were trypsinized and seeded in selection medium containing 15 μg ml$^{-1}$ Blasticidin and 250 μg ml$^{-1}$ Hygromycin. Resistant cell colonies were expanded and tested for doxycycline inducibility of the transgene. For SILAC labelling, HeLa and HCT116 cells were cultured in lysine/arginine-free DMEM (Thermo Scientific) with 10% dialysed fetal bovine serum, 2 mM L-glutamine, penicillin/streptomycin and light lysine (73 μg ml$^{-1}$) and light arginine (42 μg ml$^{-1}$). For heavy medium heavy isotope-enriched amino acids K8-lysine (L-lysine, 2HCl U-$^{13}C_6$ U-$^{15}N_2$, Cambridge Isotope Laboratories, Inc.) and R10-arginine (L-arginine U-$^{13}C_6$ U-$^{15}N_4$) were used. Cells were cultured in corresponding SILAC medium for at least seven passages and the incorporation of labelled amino acids to at least 95% was verified.

**Plasmids and bacterial strains.** All generated plasmids and Salmonella strains in this study are listed in Supplementary Table 1. S. Typhimurium strains are all in the SL1344 background. The invA deletion strain (ΔSPI1) was a kind gift of Jorge Galan and described previously[35]. The sopA deletion strain was generated using the lambda Red recombination method[36]. Herefore, lambda Red recombinase expressing temperature-sensitive plasmid pKD46 (kind gift from Dirk Bumann) was introduced into SL1344 at 30 °C. Electrocompetent SL1344 carrying pKD46 was then electroporated with PCR products harbouring the kanamycin resistance gene from pKD4 flanked by 40 bp overhangs homologous to the 5′- and 3′-ends of the sopA gene. After electroporation, bacteria were plated on kanamycin-containing lysogeny broth (LB) plates at 37 °C to prevent pKD46 propagation. Resistant clones were recovered, grown overnight at 37 °C, 180 r.p.m. and genomic DNA was isolated using GenElute Bacterial Genomic DNA Kit (Sigma Aldrich). SopA knockout was verified by PCR using primers flanking the sopA gene. For complementation of the ΔsopA strain, the sopA coding sequence and its promoter-containing upstream region encompassing 638 bp were amplified from SL1344 genomic DNA and cloned into the low-copy vector pWSK29. Complementation was performed with empty vector or with untagged or C-terminally haemagglutinin (HA)-tagged SopA WT and catalytic-dead C753A mutant. Mammalian SopA expression constructs were generated by amplifying sopA coding sequence from SL1344 genomic DNA and cloning it into pEGFP-C1 in frame with the N-terminal GFP tag. GFP-SopA WT and mutants were then transferred en bloc into pcDNA5 FRT/TO used for stable cell line generation. Plasmids for transient mammalian expression of TRIM ligases were constructed using corresponding complementary DNAs of human TRIM56 (MGC clone: 52308) and TRIM65 (MGC clone: 4385873), and cloning them into pcDNA5 FRT/TO in frame with an N-terminal FLAG tag. The pcDNA5 FRT/TO FLAG-TRIM32 construct was obtained from the University of Dundee (DU25291). TRIM39-FLAG-expressing plasmid was purchased from Hölzel. For recombinant protein production, full-length SopA and TRIM56 were cloned into pGEX-6P1 and pMAL-C2X with an N-terminal glutathione S-transferase (GST)- or maltose binding protein (MBP)-tag, respectively.

Following plasmids were made for purification of various proteins for use in in vitro ubiquitination assays, binding studies and crystallization experiments. Plasmids of mouse Ube1 and human UbcH7 in pET28a vector with C-terminal HIS tag were used for expression/purification in E. coli BL21 DE3 and subsequently used in in vitro ubiquitination assays. SopA (163–782), various constructs of TRIM56 (full-length, 1–94, 1–207 and 95–207) were cloned into a modified pET15b vector with an N-terminal HIS tag cleavable by 3C protease. The fusion construct of SopA (163–425) and TRIM56 (1–94) was generated by inserting a linker sequence coding for GSGSENLYFQGGSGS between both proteins using overlapping primers and cloned into a modified pET21a vector with C-terminal CPD-HIS tag. SopA (163–782) and TRIM56 (1–94) were also cloned into the modified pET21a vector.

All point mutations and truncations in SopA and TRIM proteins were introduced by site-directed mutagenesis according to standard protocols.

**X-ray data collection and structure determination.** Purified SopA (163–425)/TRIM56 (1–94) fusion was used at 20 mg ml$^{-1}$ in vapour diffusion crystallization trials. Crystals were grown in 28% w/v Polyacrylate 2100 sodium salt, 0.2 M NaCl and 0.1 M MES pH 6. Optimized crystals were flash frozen in liquid nitrogen. Mother liquor supplemented with 25% glycerol was used as a cryo protectant. Diffraction data were collected at Swiss Light Source (Villigen, Switzerland) and processed by XDS[37]. SopA structure (PDB code: 2QZA)[9] trimmed to contain only the residues 163–425 and TRIM32 structure (PDB code 2CT2) were used as search models in molecular replacement by PHASER[38]. To aid in model building, positions of $Zn^{2+}$ atoms were determined using ANODE[39]. Iterative model building and refinement cycles were carried out until convergence using COOT[40] and PHENIX[41], respectively. A side-by-side stereo view of the electron density covering the SopA–TRIM56 interface is provided in Supplementary Fig. 5.

**Immunoblotting and antibodies.** Proteins were separated by SDS–PAGE and transferred to nitrocellulose membranes (GE Healthcare) using wet blot transfer. Total protein was stained using Ponceau S and membranes were blocked and incubated with antibodies either in 5% milk or 5% BSA in TBS (150 mM NaCl and 20 mM Tris pH 8.0). Washes were performed in TBS-T (TBS and 0.1% Tween20).

After incubation with horesradish peroxidase-conjugated secondary antibodies, membranes were developed using chemiluminescence reagents ImmunoCruz (Santa Cruz) or TMA-6 (Lumigen). The following commercial primary antibodies were used: α-HA (16B12, Covance MMS-101, 1:1,000), α-GFP (B-2, Santa Cruz sc-9996, 1:1,000), α-FLAG (M2, Sigma F3165, 1:5,000), α-Ub (P4D1, Santa Cruz sc-8017, 1:1,000), α-TRIM56 (Abcam ab154862, 1:10,000), α-TRIM65 (Sigma HPA021578, 1:500), α-Tubulin (Sigma T9026, 1:5,000) and α-Vinculin (Sigma hVIN-1, 1:2,000). Uncropped scans of developed membranes are provided in Supplementary Figs 6,7,8 and 9 as part of the Supplementary Information.

**Salmonella infection.** Cultures of S. Typhimurium strains were inoculated from single colonies in LB with 0.3 M NaCl and grown overnight for 16 h at 37 °C with shaking 180 r.p.m. The next day, bacteria were diluted 1:33 and grown at 37 °C and 180 r.p.m. for 3 h until an optical density $OD_{600}$ of 1.8 was reached. Cell lines were grown to 80–90% confluency in DMEM + 10% FCS with penicillin and strepto-mycin. Before infection, cells were washed and incubated in antibiotics-free medium. Infections were performed for 30 min at 37 °C at a multiplicity of infection of 50 in DMEM + 10% FCS. The infection medium was aspirated and cells were incubated for 1 h in DMEM + 10% FCS containing 100 µg ml$^{-1}$ gen-tamicin to kill extracellular bacteria. For the remainder of the experiment, cells were kept in full medium with 16 µg ml$^{-1}$ gentamicin.

**Protein immunoprecipitation.** For transient co-expression experiments, $8 \times 10^5$ HEK293T cells were seeded in six-well plates. Transfection was performed 24 h later with 1 µg of corresponding plasmid DNA using GeneJuice (Novagen) according to manufacturer's instructions. Cells were harvested and lysed 24 h after transfection. For stable inducible protein expression, corresponding cells were treated with 1 µg ml$^{-1}$ doxycycline for 8 h. Cells were washed once in ice-cold PBS and subsequently lysed for 10 min on ice in immunoprecipitation (IP) lysis buffer (10 mM Tris-HCl pH 7.5, 100 mM KCl, 2 mM MgCl$_2$, 0.5% NP-40 and 1 × Pro-tease Inhibitor Cocktail EDTA-free (Roche)). Insoluble material was pelleted by centrifugation for 10 min at 15,000 r.p.m., 4 °C and supernatants were applied to lysis buffer-equilibrated antibody-coupled resin. For respective IPs, anti-FLAG (Sigma) and anti-GFP (Chromotek) matrices were used and incubated for 2 h at 4 °C with gentle rotation. Protein-bound beads were washed 5 × with IP lysis buffer and eluted by boiling in 2 × Laemmli buffer. For SILAC-coupled IP, the same amount of heavy and light labelled cells were harvested and proteins were immunoprecipitated separately. SILAC-IPs from differently labelled conditions were pooled during the last wash step and eluted together.

**Recombinant protein expression and purification.** GST- or MBP-fusion coding plasmids were transformed into E. coli BL21 (DE3). Overnight cultures were diluted 1:40 in LB medium with 200 µM ZnSO$_4$ and cells were grown to $OD_{600}$ of 0.6 at 37 °C and 140 r.p.m. Cultures were cooled down to 16 °C and protein expression was induced with 0.25 mM isopropyl-β-D-thiogalactoside. After over-night incubation (> 18 h), cells were lysed by sonication in lysis buffer (20 mM Tris-HCl pH 7.5, 150 mM NaCl, 1 mM phenylmethylsulfonyl fluoride and 0.1% β-mercaptoethanol) and incubated in lysis buffer with 0.5% Triton X-100 for 5 min on ice. Lysates were cleared by centrifugation and the supernatants were incubated with glutathione sepharose 4B beads (GE Healthcare) or amylose resin (NEB) at 4 °C with gentle rotation. Fusion protein-bound beads were washed 3 × with lysis buffer containing 0.1% Triton X-100. MBP-fusion proteins were eluted from resin in maltose-containing buffer (25 mM Tris-HCl pH 7.5 and 10 mM maltose).

C-terminal CPD-HIS-tagged fusion of TRIM56 (1–94) and SopA (163–425) was expressed in Rosetta DE3 E. coli cells. Cells were lysed in Buffer A containing 50 mM Tris pH 7.5, 10% glycerol and 300 mM NaCl. Cleared lysate was incubated with pre-equilibrated Talon beads for an hour at 4 °C. After discarding the flowthrough, the protein-bound beads were washed thoroughly with Buffer A and incubated with 100 µM of Inositol hexakisphosphate (Sigma) in Buffer A containing 50 mM NaCl, to induce the CPD self-cleavage and the release of TRIM56-SopA fusion from the beads. Protein was loaded onto anion exchange column Q-sepharose (GE Healthcare) and eluted using a linear gradient of salt from 50 mM NaCl to 1 M NaCl in Buffer A. Protein was concentrated and injected into a Superdex 200 10/300 size-exclusion column and eluted in buffer containing 10 mM HEPES pH 7.5, 150 mM NaCl and 1 mM tris(2-carboxyethyl)phosphine (TCEP). Eluted protein was buffer exchanged into 10 mM HEPES pH 7.5, 50 mM NaCl, 1 mM TCEP and concentrated to 20 mg ml$^{-1}$ using Amicon ultrafiltration device and used for crystallization. All SopA and TRIM56 constructs used in in vitro biochemical assays and isothermal titration calorimetry were purified as described above, without the ion-exchange chromatography and the final buffer exchange steps.

**Pull-down with recombinant proteins.** GST pull-down experiments were per-formed in 600 µl pull-down buffer (150 mM NaCl, 50 mM Tris pH 7.5, 5 mM dithiothreitol (DTT), 0.1% NP-40 and 0.5 mg ml$^{-1}$ BSA). GST-fusion proteins coupled to beads (5 mg) were incubated with soluble MBP-TRIM56 (500 µg) overnight at 4 °C with gentle rotation. Beads were washed with PDB, eluted with Laemmli buffer and analysed by SDS–PAGE and immunoblotting.

For Ni pull-down experiment, untagged SopA (163–782) and various constructs of HIS-tagged TRIM56 were expressed in BL21 DE3 E. coli cells. Lysates containing HIS-tagged TRIM proteins and untagged SopA were mixed and incubated with Talon beads for 1 h at 4 °C in buffer containing 50 mM Tris pH 7.5, 100 mM NaCl, 5% glycerol and 10 mM imidazole. Beads were subsequently washed and bound proteins were eluted using Laemmli buffer and analysed by SDS–PAGE.

For TUBE pull-down Salmonella-infected HeLa cells either left untreated or treated with 20 µM MG132 were harvested 30 min post infection, washed with ice-cold PBS and lysed on ice in IP lysis buffer containing the deubiquitinase inhibitor N-ethylmaleimide (10 mM Tris-HCl pH 7.5, 100 mM NaCl, 2 mM MgCl$_2$, 0.5% NP-40, 1 × Protease Inhibitor Cocktail EDTA-free (Roche) and 10 mM N-ethylmaleimide). Cell debris was pelleted by centrifugation for 10 min at 15,000 r.p.m., 4 °C and supernatants were added to GST-tagged tandem UBA (TUBE1)-coupled resin (LifeSensors). Pull-down reactions were incubated for 2 h at 4 °C with gentle rotation, beads were washed five times with PBS–Tween20 (0.1%) and bound material was eluted by boiling in Laemmli buffer.

**In vitro ubiquitination.** Unless otherwise mentioned, in vitro ubiquitination reaction mixture contained 300 nM purified Ube1 (E1), 3 µM E2 (UbcH7 or UbcH5a/b), 1 µM SopA (163–782) WT or C753A mutant and ~8 µM of various TRIM56 constructs. Ub was added to a final concentration of 25 µM and the reaction was initiated by adding 2.5 mM ATP. Ubiquitination buffer contained 50 mM Tris pH 7.5, 50 mM NaCl, 10 mM MgCl$_2$ and 0.5 mM DTT. Reaction was allowed to proceed for 1 h at 37 °C and stopped by the addition of Laemmli buffer. Samples were analysed by SDS–PAGE followed by western blotting. For the in vitro ubiquitination experiment shown in Supplementary Fig. 3e, bacterial lysates expressing WT and various mutants of TRIM56 RING domain were used instead of purified TRIM56 constructs.

**MS sample preparation.** For SILAC interactome analysis, IP eluates were sub-jected to in-gel trypsin digest[42]. To this end, eluates were run on a 10% Tris-glycine SDS–PAGE and lanes were cut into six slices each. Proteins were reduced with 10 mM DTT, alkylated with 55 mM chloroacetamide and digested overnight with 12.5 ng µl$^{-1}$ trypsin. The next day, peptides were sequentially extracted from gel pieces using 30, 80 and 100% acetonitrile-containing solvents. For proteomics and diGly proteomics experiments, SILAC labelled cells were washed twice with ice-cold PBS and lysed under denaturing conditions in 5 ml denaturation buffer (8 M Urea, 1% SDS, 50 mM Tris pH 8, 50 mM NaCl, 1 × Protease Inhibitor Cocktail EDTA-free (Roche) and 50 µM DUB inhibitor PR-619). To fragment DNA, lysates were sonicated 2 × 90 s with 1 s pulses. Protein content was measured using BCA assay (Thermo Scientific) and differentially labelled lysates were mixed in a 1:1 ratio (> 20 mg total protein). Proteins were precipitated using methanol/ chloroform and resuspended in 7 ml thiourea-containing denaturation buffer (6 M Urea, 2 M Thiourea and 50 mM Tris-HCl pH 8). Samples were reduced using 5 mM DTT and alkylated with 10 mM chloroacetamide. Proteolytic digest was performed initially with Lys-C (Wako; 1:200 substrate-enzyme ratio) for at least 4 h at room temperature. To reduce urea concentration, digests were diluted 1:5 with 25 mM Tris-HCl pH 8 and incubated for at least 16 h with trypsin (Promega; 1:200 substrate-enzyme ratio) at room temperature. Digests were stopped by addition of trifluoroacetic acid (TFA) to 0.4% and diGly proteomics samples were desalted using tC18-Sep-Paks (Whatman), eluted in 50% acetonitrile and dried by lyophilization.

**Purification of diGly-modified peptides.** After lyophilization, peptides were resuspended in 1.5 ml IAP buffer (50 mM MOPS pH 7.4, 10 mM Na$_2$HPO$_4$ and 50 mM NaCl). The pH of the peptide solution was adjusted using 1 M Tris pH 10 and insoluble material was removed (9,000 g/8 min). Supernatants were incubated with 128 µg IAP buffer equilibrated α-Lys(ε)-GG antibody coupled agarose (Cell Signaling) for 4 h at 4 °C. Beads were washed 3 × with IAP buffer and twice with MilliQ water. Bound diGly peptides were eluted 2 × with 80 µl 0.15% TFA by centrifugation and were further separated into multiple fractions using pH-based strong cation exchange chromatography[43]. Each peptide fraction was desalted using C18 material containing stage tips[44].

**Peptide identification and data analysis.** Each peptide sample was applied onto an EasyLC nano-HPLC-coupled Orbitrap Elite setup (Thermo Scientific). Peptides were injected in 0.5% acetic acid on a 15 cm-long C18-filled fused silica capillary column. Samples were separated at a flow of 200 nl min$^{-1}$ using a 226 min gra-dient of 5–33% solvent B (80% acetonitrile and 0.5% acetic acid). Peptide MS spectra were acquired in the Orbitrap ranging from $m/z$ 300 to 2,000 with a resolution of 120,000. The mass spectrometer was operated in data-dependent acquisition mode. For each cycle, the 20 peptide ions with highest intensity were sequentially isolated and CID MS/MS spectra were acquired in the Iontrap. The target values of the mass analysers were $10^6$ (Orbitrap) and $5 \times 10^3$ (Iontrap). Peptide ions with unassigned charge state and in case of diGly proteomics samples below + 3 were excluded from fragmentation. Raw data were processed with MaxQuant (1.3.0.5)[45,46] and fragment peaks were searched against human and Salmonella SL1344 UNIPROT databases. Up to two missed tryptic cleavages were allowed. Carbamidomethylation of cysteine was set as fixed modification. Variable

modifications included N-terminal acetylation, methionine oxidation and GlyGly modification. For protein quantification, a minimum of two quantified peptides was required. Both protein and peptide false discovery rate were set to 1%. Perseus was used for data sorting.

**Isothermal titration calorimetry.** Purified SopA (163–782) and TRIM56 (1–94) were buffer exchanged into 10 mM HEPES, 100 mM NaCl and 1 mM TCEP. Using VP-ITC Microcal calorimeter (GE Healthcare), 1.4 ml of 20 μM SopA was titrated with 1 mM of TRIM56 at 25 °C with 5 μl injections with a gap of 5 min between each injection. Heat of dilution was also measured by injecting 1 mM TRIM56 into the buffer and the resulting values were subtracted from the SopA–TRIM56 titration curve. Data were analysed using Origin 7 software provided by Microcal.

**Data availability.** The atomic coordinates of the SopA–TRIM56 structure have been deposited in the Protein Data Bank under accession number 5JW7. All the remaining data can be obtained from the corresponding author upon reasonable request. The following reported PDB files were used: 4AP4, 2JMD, 2QZA and 2CT2.

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

## Acknowledgements

We thank D. Bumann and B. Claudi for advice, and P. Kraiczy and V. Kempf for help with infections and all members of the Dikic lab for support and constructive discussions. We thank the Swiss Light Source staff for their support during diffraction data collection. This work was supported by grants from the DFG (SFB 1177 on selective autophagy), the SPP 1580 programme 'Intracellular compartments as places of pathogen-host-interactions', the Cluster of Excellence 'Macromolecular Complexes' of the Goethe University Frankfurt (EXC 115), LOEWE grant Ub-Net and LOEWE Centrum for Gene and Cell Therapy Frankfurt. L.H. is supported by an European Molecular Biology Organization (EMBO) long-term postdoctoral fellowship. E.F. is supported by a Boehringer Ingelheim Fonds PhD fellowship.

## Author contributions

E.F. and I.D. conceived the study and initiated the project. I.D. supervised the project. E.F. performed and analysed MS experiments, *Salmonella* infections, immunoprecipitation experiments, biochemical experiments with GST- and MBP-fusion proteins, cycloheximide chase assays, generated stable cell lines and *Salmonella sopA* knockout and

complemented strains. S.B. performed and analysed crystallography experiments including protein purification of SopA–TRIM56 fusion and HIS-tagged proteins, crystallization, structure determination, isothermal titration calorimetry, nickel pull-down experiments, modelling and *in vitro* ubiquitination experiments. L.H. and M.H. performed immunoprecipitation and infection experiments. S.K. performed *in vitro* ubiquitination experiments. E.F., S.B. and I.D. wrote and all authors edited the manuscript.

## Additional information

**Competing financial interests:** The authors declare no competing financial interests.

**Publisher's note**: 

