## [Peer Review File · Nature Communications]

Reviewers' comments:

Reviewer #1 (Remarks to the Author):

A. The authors report finding 2 TRIM proteins as the main interactors of the bacterial virulence factor SopA. They go on to determine the structures of complexes of truncations of SopA and TRIM 56 in order to work out how host TRIM proteins are targeted by SopA.

B. The results are original and interesting, if slightly overstated in the abstract. I would advise the authors to focus on their mechanistic findings, which are novel and interesting, as opposed to the claim that 'here we demonstrate that bacterial virulence factor SopA is a HECT E3-ligase that targets host RING E3 ligases TRIM56 and TRIM65 during infection'.

In my opinion, this sentence is misleading and suggests that these authors discovered that SopA is a HECT E3 ligase (recognised elsewhere in the manuscript and correctly credited as reference 19), and that they discovered it targets TRIM proteins, also found by someone else (Kamanova et al., ref 24). The elucidation of the mechanism is significant enough without needing to appropriate the findings of others. I would recommend changing the abstract to reflect this - perhaps something along the lines of 'SopA is a HECT-like E3 ligase that targets host ligases. Here we demonstrate that specifically TRIMs 56 and 65 are the host proteins targeted for degradation...'.

C-G. Overall the work is well-presented, and easy to follow. The data are high quality, and I have no concerns regarding the robustness or validity of the data and conclusions, it is an excellent piece of work. There are however a number of issues that need to be addressed prior to publication:

Lack of attention to detail with the description of the work. The majority of the paper discusses the structure of the TRIM/SopA complex and the insights derived from it. Yet the table that contains the data used to determine the structure doesn't follow even basic scientific conventions - there are very long significant figures, and no units for the Wilson or average B-factors, of bond lengths, or angles.

The authors say they fused the proteins with a flexible linker - what was the rationale for this? Have they encountered anybody else's work with RING proteins that have required this methodology? Do they anticipate any interference or caveat with the introduction of the linker? There is no description in the main manuscript of the purification and expression of the proteins. I cannot see why figure 3 has a cartoon of 2 blobs plus a sequence of a linker, but there's not description of the linker and construct in the materials and methods. I do not think figure 3a needs to be in the main manuscript.

Similar to this, I am unclear why the authors choose to put indirect IP binding data in the main figure 2, but the physical measurements of interaction as supplementary. The authors could consider moving sup figs 2a and 2b to the main figure 2.

Some of the reporting of the crystallographic findings is a bit overplayed - 756 squared angstroms buried surface area seems unrealistically precise at 2.9 Å resolution. I think 700 or 750 would be more appropriate but certainly not nearest angstrom.

In my copy of text, the alpha carbons come out as 'a'.

How does the view in figure 3c relate to figure 3b? It is difficult to appreciate the relationship between the orientations for a non-structural biologist. In fig 3c, since there is a lot of discussion in the results about the polar contacts, some drawing of polar contacts such as hydrogen bonds would be useful, or

a stereo image of the interface. It is difficult to discern from the 2 dimensional image what is close to what or making interactions with what.

On page 5, results, bottom, transiently expressed GFP-SopA is referred to as wild-type, but it is not wild-type because it has a GFP tag fused to it.

S. Typhimurium (top of page 6) should be *S. typhimurium* and in italics.

p.6 line 143 - remove 'possibly' - already covered by 'whether'.

p10. What does 'most prominent' mean (line 251)?

The materials and methods in the main manuscript do not include details of the purification of the proteins used in experiments. Neither is there any detail of the assays, IPs, or activity assays, what amounts of each enzyme were used, how they were made, what the purity is. The only experimental procedure that seems to have been described in the main manuscript is the proteomics data, and I don't follow the logic of putting so much of the biochemistry and biophysics in supplementary, given the focus of the manuscript. In addition, this is to be an online only publication, can't the methods be included? Or make all of them supplementary.

G. Unless there is no publication associated with a structure, every PDB file used for analysis and mentioned should also reference the publication that described it. These are original findings made by fellow professionals. 4AP4 is from Plechananova et al., (cited, but not with the PDB code) and 2JMD is from Mercier et al., 2007, Protein Science, not cited.

H. My only comments regarding clarity would be

1) to have the paper edited for use of English (for example, on page 7 it is stated that the linker residues between TRIM56 and SopA including the 17 residues from the N-terminus of TRIM56 are disordered....however, the linker does not 'include' the 17 residues from the n-terminus of trim56, so I think the authors mean linker residues are missing. 17 N-terminal residues are also missing.). Some other examples: p8 the zinc binding loop fits 'agreeably' into the cavity, I am not sure what is intended by agreeably, it seems to be a value judgement! Perhaps the authors simply mean that it fits without clashes. The next sentence says the RINGs appear to be clashing heavily - again, this could be reduced to 'clash'.

p8. Leu25 and Glu26 are fairly conserved - again, this is a vague term. The Leucine is almost completely conserved with 4 exceptions, while the glutamate is not a glutamate as often as it is - this equates to 75% conservation of Leu, and 50% of the Glu, in this alignment.

2) Consider integrating the introduction or reversing the order of topics. At the moment it reads like another 'mechanism of E3 ligases' paper for the first page, and then goes into a specific biological question. In my opinion, the excitement and novelty of the work would be better served by introducing the biological problem first, and the mechanistic insights second. Up to you!

Reviewer #2 (Remarks to the Author):

In this manuscript, Fiskin et al employ a proteomics approach to identify two host TRIMs as novel targets of the *S. Typhi* HECT E3 ligase SopA. They proceed to perform detailed functional and mechanistic characterisation of the interaction between these proteins, including detailed structural

studies and analysis of the domains of each protein necessary for interaction. They complete their study by demonstrating that SopA has a dual mechanism of action to inhibit TRIM56/65, both by enhancing their ubiquitination and proteasomal degradation, and by blocking interaction of TRIM56 via occlusion of the E2 interacting surface.

This study was very well performed, well written and will be of significant interest and value to the field. I have no major concerns and believe relatively little needs to be modified prior to publication. Most of the concerns below could be addressed via modifications to the text:

1. The authors comment that 'no other TRIM protein was found significantly enriched in our SopA interactome studies', and '... SopA specifically interacts with the RING domains of TRIM56 and TRIM65 amongst ~600 RING domain containing human proteins'. However, in addition to TRIMs56/65, they only actually detect 3-9 other TRIM proteins in their Figure 1 IPs, with H:L or L:H ratios ~1 (i.e. not enriched). It is therefore difficult to know how many other TRIM proteins might have been enriched, but failed to be identified due to low abundance (or lack of expression by the cell lines employed) / lack of suitable peptides etc - this should be clarified. Furthermore, the authors (a) should include the number of unique / razor peptides for each protein identified in the supplementary tables (for all proteomics experiments), (b) could usefully label SopA in figure 1d (I note it was only enriched in 1 of 2 replicates); and (c) could comment on the proteins with negative ratios in the 'forward' experiment, a number of which are contaminants. The final proteomics experiment (Figure 6b) was not conducted as a 'label-swap'; this is fine as a biological replicate, however the authors could comment on the proteins (HSDL1) etc that appear to exhibit decreased ubiquitination in the presence of SopA (or increased ubiquitination in the presence of the SopA deletion mutant).

2. Experiment 1c - the 3rd lane in 'Input' / 'IP:GFP' should be labelled GFP-SopA C753A

3. The authors make a structural predictions regarding the interactions of the TRIMs with SopA that are not fully validated. For example, (a) the positioning of the central α -helix of the RING domains relative to the first Zn-binding loop is predicted to be crucial in defining the specificity of SopA. This is derived from examining predicted structures of a small number of other TRIMs in comparison to TRIM56, which they verify do not interact with SopA. (b) TRAF6 is predicted not to interact with SopA, as it contains a hydrophobic Met75 rather than Glu25 in TRIM56. Could this be addressed by substitution?

4. In figures 6c and 7a, the effect of reconstituting SopA in their SopA deletion mutant seems to be more potent than wt SL1344 (both in terms of ubiquitination of TRIM56 (experiment 6C, and degradation of TRIM56 (experiment 7a). Why is this - was the level of SopA expression examined in each case?

Reviewer #3 (Remarks to the Author):

Overall evaluation

The article by Fiskin et al. investigates the function of the *S. enterica* T3SS-1 effector protein SopA using a structure-guided approach. While the structural analysis generates appealing images, the amount of new information gained from this analysis is rather limited (point 2) and the majority of the conclusions are confirmatory (point 1).

Major points

1) The author's state in their abstract that they "demonstrate that the bacterial virulence factor SopA is a HECT-like E3 ligase that targets host RING E3 ligases TRIM56 and TRIM65 during infection" and that "SopA HECT domain mediates ubiquitination of TRIM56 and TRIM65 resulting in their proteasomal degradation during bacterial infection". However, the fact that SopA contributes to the stimulation of innate immune responses by binding two host E3 ubiquitin ligases, TRIM56 and TRIM65, which results in their increased ubiquitination, has already been demonstrated by Jorge Galan's group in reference 24. In other words, two of the main findings reported in the abstract are confirmatory.

2) The amount of new information provided in the abstract is limited to insights gleaned from the crystal structure, which revealed "SopA inhibits the E3 ligase activity of TRIM56 by occluding the E2 interacting surface of TRIM56". This finding in itself does not represent a large advance. Since SopA-mediated ubiquitination leads to TRIM56 degradation, the question arises whether masking a binding surface of a degrading protein has biological significance. No follow-up studies were performed to investigate the biological significance of SopA-mediated masking of the TRIM56 E2-binding surface. Thus, the only novel observation reported in this manuscript is preliminary and of questionable biological relevance.

Specific points

3) Lines 111-117: This information is redundant with the abstract and should be removed.

Answers to reviewer's comments:

Reviewer #1 (Remarks to the Author):

A. The authors report finding 2 TRIM proteins as the main interactors of the bacterial virulence factor SopA. They go on to determine the structures of complexes of truncations of SopA and TRIM 56 in order to work out how host TRIM proteins are targeted by SopA.

B. The results are original and interesting, if slightly overstated in the abstract. I would advise the authors to focus on their mechanistic findings, which are novel and interesting, as opposed to the claim that 'here we demonstrate that bacterial virulence factor SopA is a HECT E3-ligase that targets host RING E3 ligases TRIM56 and TRIM65 during infection'.

In my opinion, this sentence is misleading and suggests that these authors discovered that SopA is a HECT E3 ligase (recognised elsewhere in the manuscript and correctly credited as reference 19), and that they discovered it targets TRIM proteins, also found by someone else (Kamanova et al., ref 24). The elucidation of the mechanism is significant enough without needing to appropriate the findings of others. I would recommend changing the abstract to reflect this - perhaps something along the lines of 'SopA is a HECT-like E3 ligase that targets host ligases. Here we demonstrate that specifically TRIMs 56 and 65 are the host proteins targeted for degradation...!'

Ans: We understand the reviewer's comment. We now changed the abstract to correctly reflect the existing literature and our findings.

C-G. Overall the work is well-presented, and easy to follow. The data are high quality, and I have no concerns regarding the robustness or validity of the data and conclusions, it is an excellent piece of work. There are however a number of issues that need to be addressed prior to publication:

Lack of attention to detail with the description of the work. The majority of the paper discusses the structure of the TRIM/SopA complex and the insights derived from it. Yet the table that contains the data used to determine the structure doesn't follow even basic scientific conventions - there are very long significant figures, and no units for the Wilson or average B-factors, of bond lengths, or angles.

Ans: The table should have contained the units. We thank the reviewer for pointing this out. We updated the crystallography table to include the units for various parameters.

The authors say they fused the proteins with a flexible linker - what was the rationale for this? Have they encountered anybody else's work with RING proteins that have required this methodology? Do they anticipate any interference or caveat with the introduction of the linker? There is no description in the main manuscript of the purification and expression of the proteins. I cannot see why figure 3 has a cartoon of 2 blobs plus a sequence of a linker, but there's not description of the linker and construct in the materials and methods. I do not think figure 3a needs to be in the main manuscript.

Ans: We now clearly state the rationale for using the fusion construct (Main text, page 6, line 155-157). The details of the construct are also described in the materials and methods section (Main text, Page 16, line 414-417). We moved Figure 3a from the main manuscript to supplementary Figure 2b. SopA and TRIM56 interact transiently during the ubiquitination. In order to stabilize the complex for crystallization we fused them. Fusing two different proteins increases the local concentration of two molecules in solution and effectively supports the low affinity protein complex formation (please see PMID: 23225024 for review). This technique is not specific to RING proteins and has been successfully applied in other instances such as

interactions between LC3/GABARAP modifiers with LC3-interacting regions (PMID: 25498145; PMID: 23805866). Our mutational analysis in solution provides clear evidence that the linker does not interfere with correct SopA-TRIM56 complex formation.

Similar to this, I am unclear why the authors choose to put indirect IP binding data in the main figure 2, but the physical measurements of interaction as supplementary. The authors could consider moving sup figs 2a and 2b to the main figure 2.

Ans: We thank the reviewer for this suggestion. We added the ITC measurements of the binding constant to main Figure 2 as panel g.

Some of the reporting of the crystallographic findings is a bit overplayed - 756 squared angstroms buried surface area seems unrealistically precise at 2.9 Å resolution. I think 700 or 750 would be more appropriate but certainly not nearest angstrom.

Ans: We agree that at 2.9Å, measurements of buried surface area at protein interfaces are better represented as round figures. We now changed that number from 756 to 750.

In my copy of text, the alpha carbons come out as 'a'.

Ans: This has been changed.

How does the view in figure 3c relate to figure 3b? It is difficult to appreciate the relationship between the orientations for a non-structural biologist. In fig 3c, since there is a lot of discussion in the results about the polar contacts, some drawing of polar contacts such as hydrogen bonds would be useful, or a stereo image of the interface. It is difficult to discern from the 2 dimensional image what is close to what or making interactions with what.

Ans: We appreciated the reviewer's comments about improving Figure 3. We included rotation angles between shown conformations to specify relationships between them in the figure legend. We introduced hydrogen bond drawing (also the distance) into the figure showing the polar contacts.

On page 5, results, bottom, transiently expressed GFP-SopA is referred to as wild-type, but it is not wild-type because it has a GFP tag fused to it.

Ans: This has been changed.

S. Typhimurium (top of page 6) should be *S. typhimurium* and in italics.

Ans: We thank the reviewer for pointing out this mistake, we corrected it from 'S.Typhimurium' to 'S. Typhimurium', to our knowledge the latter represents the correct notation as 'Typhimurium' denotes a serovar (not a species) it is capitalized.

p.6 line 143 - remove 'possibly' - already covered by 'whether'.

Ans: We removed 'possibly'.

p10. What does 'most prominent' mean (line 251)?

Ans: We changed wording from 'most prominent' to 'with the highest SILAC ratios of all quantified peptides'.

The materials and methods in the main manuscript do not include details of the purification of the proteins used in experiments. Neither is there any detail of the assays, IPs, or activity

assays, what amounts of each enzyme were used, how they were made, what the purity is. The only experimental procedure that seems to have been described in the main manuscript is the proteomics data, and I don't follow the logic of putting so much of the biochemistry and biophysics in supplementary, given the focus of the manuscript. In addition, this is to be an online only publication, can't the methods be included? Or make all of them supplementary.

Ans: We now moved the complete material and methods section from the supplementary information to the main manuscript.

G. Unless there is no publication associated with a structure, every PDB file used for analysis and mentioned should also reference the publication that described it. These are original findings made by fellow professionals. 4AP4 is from Plechananova et al., (cited, but not with the PDB code) and 2JMD is from Mercier et al., 2007, Protein Science, not cited.

Ans: We regret these mistakes. Articles describing these structures are now properly cited.

H. My only comments regarding clarity would be

1) to have the paper edited for use of English (for example, on page 7 it is stated that the linker residues between TRIM56 and SopA including the 17 residues from the N-terminus of TRIM56 are disordered....however, the linker does not 'include' the 17 residues from the n-terminus of trim56, so I think the authors mean linker residues are missing. 17 N-terminal residues are also missing.). Some other examples: p8 the zinc binding loop fits 'agreeably' into the cavity, I am not sure what is intended by agreeably, it seems to be a value judgement! Perhaps the authors simply mean that it fits without clashes. The next sentence says the RINGs appear to be clashing heavily - again, this could be reduced to 'clash'.

Ans: We take note of the reviewer's comments regarding language. We changed the mentioned passages accordingly. We proofread and adjusted the paper.

p8. Leu25 and Glu26 are fairly conserved - again, this in a vague term. The Leucine is almost completely conserved with 4 exceptions, while the glutamate is not a glutamate as often as it is - this equates to 75% conservation of Leu, and 50% of the Glu, in this alignment.

Ans: We changed the wording accordingly.

2) Consider integrating the introduction or reversing the order of topics. At the moment it reads like another 'mechanism of E3 ligases' paper for the first page, and then goes into a specific biological question. In my opinion, the excitement and novelty of the work would be better served by introducing the biological problem first, and the mechanistic insights second. Up to you!

Ans: We thank the reviewer for this comment. As suggested, we now modified the introduction and introduce the biological problem before the mechanistic insights.

Reviewer #2 (Remarks to the Author):

In this manuscript, Fiskin et al employ a proteomics approach to identify two host TRIMs as novel targets of the S. Typhi HECT E3 ligase SopA. They proceed to perform detailed functional and mechanistic characterisation of the interaction between these proteins, including detailed structural studies and analysis of the domains of each protein necessary for interaction. They complete their study by demonstrating that SopA has a dual mechanism of action to inhibit TRIM56/65, both by enhancing their ubiquitination and proteasomal

degradation, and by blocking interaction of TRIM56 via occlusion of the E2 interacting surface.

This study was very well performed, well written and will be of significant interest and value to the field. I have no major concerns and believe relatively little needs to be modified prior to publication. Most of the concerns below could be addressed via modifications to the text:

1. The authors comment that 'no other TRIM protein was found significantly enriched in our SopA interactome studies', and '... SopA specifically interacts with the RING domains of TRIM56 and TRIM65 amongst ~600 RING domain containing human proteins'. However, in addition to TRIMs56/65, they only actually detect 3-9 other TRIM proteins in their Figure 1 IPs, with H:L or L:H ratios ~1 (i.e. not enriched). It is therefore difficult to know how many other TRIM proteins might have been enriched, but failed to be identified due to low abundance (or lack of expression by the cell lines employed) / lack of suitable peptides etc - this should be clarified.

Ans: We thank the reviewer for these comments. To provide a more complete view of TRIM proteins detected in our experiments, we now added supplementary Figures 4c and 4d containing the SILAC ratios and peptide intensities of TRIM proteins, which have been identified in our interactome and ubiquitinome experiments, but were not enriched/regulated by SopA. Given that not all annotated TRIM proteins were detected in these experiments, we changed the wording in the manuscript to reflect this fact. We changed the above-mentioned sentences to: 'no other detected TRIM protein was found significantly enriched...' and 'SopA specifically interacts with the RING domains of TRIM56 and TRIM65 amongst a large number of expressed RING domain...'

Furthermore, the authors (a) should include the number of unique / razor peptides for each protein identified in the supplementary tables (for all proteomics experiments)

Ans: The razor/unique peptides are now included in Table S1 and S2.

(b) could usefully label SopA in figure 1d (I note it was only enriched in 1 of 2 replicates);

Ans: We now marked SopA in Figure 1d. The reason for no apparent SopA enrichment in the forward interactome experiment using infected cells is due to the lack of heavy SILAC labeling of proteins synthesized by Salmonella. During our infection experiments only mammalian host cells are heavy labeled, whereas added bacteria are non-labeled (light) and do not incorporate heavy amino acids in the analysed time frame.

and (c) could comment on the proteins with negative ratios in the 'forward' experiment, a number of which are contaminants.

Ans: We thank the reviewer for this comment. The figure legends for figures 1b and 1d now state that the mentioned proteins with strong negative ratio in the forward experiments (upper left quadrant) represent expected contaminants. These include cell culture medium-derived BSA or proteins highly expressed in skin tissue such as Dermcidin, Desmoglein, Junction plakoglobin and Keratinocyte proline-rich protein likely derived from sample handling.

The final proteomics experiment (Figure 6b) was not conducted as a 'label-swap'; this is fine as a biological replicate, however the authors could comment on the proteins (HSDL1) etc that appear to exhibit decreased ubiquitination in the presence of SopA (or increased ubiquitination in the presence of the SopA deletion mutant).

Ans: We think that there could be multiple explanations for proteins appearing to exhibit decreased ubiquitination in the presence of SopA. (1) The decreased ubiquitination of

proteins such as HSDL1 might be the result of TRIM56/65 degradation by SopA, in particular HSDL1 or others might represent TRIM56/65 substrates. (2) On the other hand it is also possible that apparent abundance changes of identified diGly sites such as HSDL1 K109 in fact arise from incorrect SILAC ratio quantification. The latter can be addressed in part through label swap experiments.

2. Experiment 1c - the 3rd lane in 'Input' / 'IP:GFP' should be labelled GFP-SopA C753A

Ans: The labeling is correct, these lanes are Input and IP:GFP of GFP-SopA, the second and third lanes represent the same condition.

3. The authors make a structural predictions regarding the interactions of the TRIMs with SopA that are not fully validated. For example, (a) the positioning of the central α -helix of the RING domains relative to the first Zn-binding loop is predicted to be crucial in defining the specificity of SopA. This is derived from examining predicted structures of a small number of other TRIMs in comparison to TRIM56, which they verify do not interact with SopA. (b) TRAF6 is predicted not to interact with SopA, as it contains a hydrophobic Met75 rather than Glu25 in TRIM56. Could this be addressed by substitution?

Ans: We tested binding of TRAF6 M75E, M75D and M75A mutants to GFP-SopA C753A using co-immunoprecipitation experiments (see figure for reviewer). Whereas the used controls (TRIM56 and TRIM65 full length and deltaC constructs) bound SopA, we did not observe any SopA interaction for TRAF6 WT or M75E, M75D and M75A mutant proteins. This result suggests that there are additional features preventing TRAF6-SopA interaction. Indeed, our model of the TRAF6 RING in complex with SopA (Figure 4c) still suggests the presence of sterical clashes even though they are less pronounced compared to the other modeled RING domains. As Met75Glu mutation is not sufficient to induce interaction of TRAF6 RING domain to SopA, we removed the sentence on TRAF6 Met75 from the manuscript (Main text, page 8, line 212).

4. In figures 6c and 7a, the effect of reconstituting SopA in their SopA deletion mutant seems to be more potent than wt SL1344 (both in terms of ubiquitination of TRIM56 (experiment 6C, and degradation of TRIM56 (experiment 7a). Why is this - was the level of SopA expression examined in each case?

Ans: Despite being driven by its endogenous promoter, the reconstituted strains (deltaSopA+SopA WT or C753A) express SopA from a plasmid (pWSK29) and not from the bacterial genome. Plasmid-based expression results in elevated SopA levels in these strains. We quantified SopA protein levels in the samples shown in Figure 6c using proteomics and label-free quantification and included these results now in the new Supplementary Figure 3g and in the manuscript (Main text, Page 10, line 250-252). Our proteomic analysis revealed ~35 fold higher SopA peptide intensities in cells infected with SopA reconstituted deletion strains (deltaSopA+SopA WT or C753A) compared to cells infected with SL1344 WT (endogenous SopA). Importantly, this is not the case for other effectors SopE and SopB as both display similar levels under these conditions. Increased modification and degradation of TRIM56 (Figure 6c and Figure 7a) in deltaSopA+SopA WT or C753A infection is therefore a result of elevated SopA amounts under these conditions.

Reviewer #3 (Remarks to the Author):

Overall evaluation

The article by Fiskin et al. investigates the function of the *S. enterica* T3SS-1 effector protein SopA using a structure-guided approach. While the structural analysis generates appealing images, the amount of new information gained from this analysis is rather limited (point 2)

and the majority of the conclusions are confirmatory (point 1).

Major points

1) The author's state in their abstract that they "demonstrate that the bacterial virulence factor SopA is a HECT-like E3 ligase that targets host RING E3 ligases TRIM56 and TRIM65 during infection" and that "SopA HECT domain mediates ubiquitination of TRIM56 and TRIM65 resulting in their proteasomal degradation during bacterial infection". However, the fact that SopA contributes to the stimulation of innate immune responses by binding two host E3 ubiquitin ligases, TRIM56 and TRIM65, which results in their increased ubiquitination, has already been demonstrated by Jorge Galan's group in reference 24. In other words, two of the main findings reported in the abstract are confirmatory.

2) The amount of new information provided in the abstract is limited to insights gleaned from the crystal structure, which revealed "SopA inhibits the E3 ligase activity of TRIM56 by occluding the E2 interacting surface of TRIM56". This finding in itself does not represent a large advance. Since SopA-mediated ubiquitination leads to TRIM56 degradation, the question arises whether masking a binding surface of a degrading protein has biological significance. No follow-up studies were performed to investigate the biological significance of SopA-mediated masking of the TRIM56 E2-binding surface. Thus, the only novel observation reported in this manuscript is preliminary and of questionable biological relevance.

Ans: There are multiple novel findings reported in our story. In addition to providing high quality mass-spectrometry and biochemistry data regarding the binding of SopA and TRIM proteins, we showed that SopA specifically assembles K48- and K11-linked ubiquitin chains on TRIMs and targets them for degradation. We demonstrate that the binding is mediated through the N-terminal regions of both SopA and TRIM proteins. We also determined the crystal structure of SopA in complex with TRIM56 adding to only a handful of ubiquitin ligase-substrate structures in the Protein Data Bank. Apart from revealing a common mode of recognition by SopA towards TRIM56 and TRIM65 our structure also provided a basis for the selective nature of SopA towards TRIM56 and TRIM65. Regarding SopA masking the E2-binding surface of the TRIM56 RING domain, while we do state in the paper that such an inhibition mechanism during infection is subject to many parameters, we think that such binding of SopA to the TRIM56 RING domain makes a strong case for a clever way of ubiquitinating an E3 ligase while avoiding getting ubiquitinated by it. Kamanova et al. and our manuscript disagree on the fundamental nature of SopA targeting TRIM56 and TRIM65. In Kamanova et al., the authors concluded that SopA does not cause degradation of TRIMs, while we provide compelling evidence of in vitro, cell based and bacterial infection data showing that these ubiquitination events are degradative. Therefore, we think that our findings elucidate the mode of regulation of these TRIM proteins by SopA.

Specific points

3) Lines 111-117: This information is redundant with the abstract and should be removed.

Ans: This information is modified now in the introduction.

[Redacted] confidential for reviewer only

Reviewers' comments:

Reviewer #1 (Remarks to the Author):

All my concerns have been adequately addressed.

Reviewer #2 (Remarks to the Author):

I am happy with the responses and revisions and would now consider the manuscript suitable for publication.

Reviewer #3 (Remarks to the Author):

The article by Fiskin et al. investigates the function of the *S. enterica* T3SS-1 effector protein SopA using a structure-guided approach. The main novel findings are that SopA inhibits TRIM56 and TRIM65 function by blocking E3 ligase activity and targeting TRIM56 and TRIM65 for degradation by specifically assembling K48- and K11-linked ubiquitin chains on TRIMs. These findings directly contradict earlier work by another group, which should be stated clearly throughout the manuscript to highlight the novelty of the study (point 1) and needs to be confirmed by additional functional tests (point 2).

1) Jorge Galan's group proposed that SopA activates TRIM56 and TRIM65 to trigger RIG-I and MDA5 signaling, while the author's data suggest that SopA does the exact opposite. This needs to be clearly stated in the abstract and results sections, while the discussion should include some dialogue about possible explanations for the discrepant results.

2) Previous work suggests that SopA-mediated activation of TRIM65 results in interaction with MDA5 thereby enhancing the expression of beta-interferon in HEK293 cells. The author's work suggests that SopA inhibits TRIM65, which would prevent or lower beta-interferon production in host cells. This prediction needs to be tested.

Answers to reviewer's comments:

Reviewer #1 (Remarks to the Author):

All my concerns have been adequately addressed.

Reviewer #2 (Remarks to the Author):

I am happy with the responses and revisions and would now consider the manuscript suitable for publication.

Reviewer #3 (Remarks to the Author):

The article by Fiskin et al. investigates the function of the *S. enterica* T3SS-1 effector protein SopA using a structure-guided approach. The main novel findings are that SopA inhibits TRIM56 and TRIM65 function by blocking E3 ligase activity and targeting TRIM56 and TRIM65 for degradation by specifically assembling K48- and K11-linked ubiquitin chains on TRIMs. These findings directly contradict earlier work by another group, which should be stated clearly throughout the manuscript to highlight the novelty of the study (point 1) and needs to be confirmed by additional functional tests (point 2).

1) Jorge Galan's group proposed that SopA activates TRIM56 and TRIM65 to trigger RIG-I and MDA5 signaling, while the author's data suggest that SopA does the exact opposite. This needs to be clearly stated in the abstract and results sections, while the discussion should include some dialogue about possible explanations for the discrepant results.

Ans: We now adjusted abstract (Main text, page 2, line 32) and results section (Main text, page 9, line 231 and page 11, line 287) of the manuscript stating the respective findings of the study by the Galan group. Furthermore, we expanded the discussion and possible explanations for the discrepancy between our present findings and their reported data (Main text, page 13, line 348-368).

2) Previous work suggests that SopA-mediated activation of TRIM65 results in interaction with MDA5 thereby enhancing the expression of beta-interferon in HEK293 cells. The author's work suggests that SopA inhibits TRIM65, which would prevent or lower beta-interferon production in host cells. This prediction needs to be tested.

Ans: We tested the effect of SopA WT and C753A on the expression of interferon- β in HEK293 cells expressing MDA5 and TRIM65 using an interferon- β driven luciferase reporter (see attached figures 1 and 2 for reviewer). In our experiments we did not detect stimulation of type I interferon expression by SopA and instead observed SopA ligase activity to confer decreased interferon- β transcription concurrent with a decrease in TRIM65 protein levels.

We have thus stated in the discussion:

Importantly, we did not detect stimulation of TRIM56 or TRIM65-mediated type I interferon expression by SopA, but observed that SopA-activity confers decreased interferon- β transcription instead. While it can not be excluded that the relative expression levels and E3 ligase activities of secreted SopA and host TRIM56 and TRIM65 in vivo determine the outcome of this host-pathogen targeting event and account for the discrepancy in presented findings and previous data, our observations in cultured epithelial cells indicate a progressive decay of endogenous TRIM56 and TRIM65 protein levels upon standard Salmonella infection conditions.

[Redacted] confidential for reviewer only

[Redacted] confidential for reviewer only

REVIEWERS' COMMENTS:

Reviewer #3 (Remarks to the Author):

The authors have addressed my concerns and present a compelling set of data supporting a role of SopA that contradicts previous findings. This represents an important contribution that will move the field forward.